# Endothelin-3 and T-type Ca²⁺ channels drive enteric neural crest cell calcium activity, contractility and migration

Nicolas R. Chevalier [1] ✉, Fanny Gayda[2], Nadège Bondurand[2], Ze Chi Chan[1], Thierry Savy [1], Monique Frain[1], Amira El Merhie [1], Lenuta Canta[3], Monica Dicu[3], Isabelle Le Parco[4] & Léna Zig[1]

Enteric neural crest cells (ENCCs) colonize the gut during embryogenesis and migration defects give rise to Hirschsprung disease (HD). Mutations in GDNF/RET and EDN3/EDNRB are known to be causal in HD. Here, we show that migrating ENCCs in mice exhibit endogenous EDN3/EDNRB-gated calcium activity, mediated by chloride channels, T-type Ca²⁺ channels and inositol trisphosphate-sensitive intracellular-store release. We find that inhibiting Ca²⁺ activity results in ENCC migration defects, while exciting it promotes migration by increasing ENCC contractility and traction force to the extracellular matrix. Our study demonstrates that embryonic endothelin-mediated neural crest migration and adult endothelin-mediated vasoconstriction is one and the same phenomenon, taking place in different cell types. Our results suggest a functional link between rare mutations of *CACNA1H* (the gene encoding CaV3.2) and HD, and pave the way for understanding neurocristopathies in terms of neural crest cell bioelectric activity deficits.

Neural crest cells (NCC) are a population of multipotent, highly migratory cells common to all vertebrates. They colonize developing organs during embryonic development to give rise to a variety of structures including cranio-facial bone, chromaffin cells of the adrenal medulla, Schwann cells of the peripheral nervous system and the neurons and glia of the enteric nervous system (ENS)[1]. NCCs have attracted much scientific attention because of their paramount contributions to vertebrate physiological and pathological[2,3] development, to domestication-induced phenotypes[4,5] and because aberrantly expressed NCC developmental programs can lead to cancer tumor invasiveness[6,7]. Vagal-derived[8] enteric neural crest cells (ENCCs) invade the gut rostro-caudally between 9.5 and 13.5 days of development in mice to form the ENS. ENCC migration defects result in colonic aganglionosis at birth, a syndrome known as Hirschsprung disease (HD). HD is one of the most frequent neurocristopathies, affecting ~1:5000 births[9]. The mutations hitherto identified, mostly in the GDNF/RET[10–12] and EDN3/EDNRB[13,14] signaling cascades, are of incomplete penetrance and account for only 50 % of the familial and 15-20 % of the sporadic cases[15]: the cause of most HD cases is today not understood. Bioelectric activity[16–18], as measured by Ca²⁺ imaging or membrane potential changes, is increasingly recognized as a key player of stem cell[16–18] and NCC[19–22] behavior. Spontaneous propagating Ca²⁺ waves in ENCCs[23] were found to depend on purinergic signaling, but their implication for migration and HD remained uncertain. Here, we show that most of the Ca²⁺ activity in ENCCs occurs as unsynchronized single-cell events and not as waves, that ENCC Ca²⁺ activity is driven by EDN3/EDNRB, via the opening of Cl⁻ and T-type Ca²⁺ channels, and that Ca²⁺ activity is intimately correlated with ENCC migration potential through a simple mechanism: elevated intracellular Ca²⁺ leads to increased ENCC contractility and traction force to the extracellular matrix that allows them to migrate down the gut. This mechanism renews our understanding of how EDN3/EDNRB affects ENCC migration. Disruption of any components of this mechanism can result in a NCC migration defect,

[1]Laboratoire Matière et Systèmes Complexes, Université Paris Cité, CNRS UMR 7057, 10 rue Alice Domon et Léonie Duquet, Paris, France. [2]Laboratory of Genetics of Developmental Disorders, Imagine Institute, INSERM UMR 1163, Université Paris Cité, 24 Boulevard du Montparnasse, Paris, France. [3]UNIIVO, 1 cour du Havre, Paris, France. [4]Institut Jacques Monod, Université Paris Cité, CNRS, Paris, France. ✉e-mail: nicolas.chevalier@u-paris.fr

opening-up new perspectives for neurocristopathy and collective cell migration research.

## Results

### Enteric neural crest cells display spontaneous Ca²⁺ activity during gut invasion

We monitored Ca²⁺ activity (CA) on ex-vivo mouse embryonic guts expressing the intracellular Ca²⁺ reporter GCaMP6f specifically in NCCs (Fig. 1a, Supplementary Movie 1), and measured its spatial distribution and characteristics with an automated analysis pipeline (Fig. S1). CA in migrating ENCCs was spontaneous and occurred mostly as asynchronous, non-propagating, single-cell events. Ca²⁺ transients could sometimes propagate radially to neighboring cells, but such wave-like events[23] occurred, at E11.5, at an activity of $CA_{wave} = 7.5 \times 10^{-4} \pm 2 \times 10^{-4}$ events/min/100 µm² ($\pm$ SEM, $n = 15$), i.e. ~100-1000 times less frequently than single-cell events (Fig. 1b). Endogenous CA at the ENCC migration front decreased from E10.5 to E12.5 (Fig. 1b, Fig. S2, Supplementary Movie 1). At E11.5, it was significantly higher at the ENCC wavefront located at the ileo-cecal junction (ilcc) than in trailing ENCCs (Fig. 1c), concentrating at the cecum and antimesenteric ileum border (Figs. 1a and S3). CA was also present in transmesenteric ENCCs (Fig. S3). CA was insensitive to tetrodotoxin (Figs. 1d and S4), and therefore did not stem from neural activity; the latter could however be elicited by veratridine in proximal gut segments (Figs. 1e and S4). Endogenous CA co-localized with the ENCC-specific marker Sox10, but only very partially with the neuronal marker Tuj1 (Supplementary Movie 2). These results indicate that spontaneous CA originated in ENCCs and was most intense during early gut colonization stages, decreasing as ENCCs gradually differentiated to neurons and glia.

### Endogenous Ca²⁺ activity is induced by EDN3/EDNRB

We next investigated whether the ligand/receptor pair EDN3/EDNRB involved in ENCC development was related to ENCC electric activity. Application of 1 nM EDN3 at E11.5 led to a threefold CA increase (Supplementary Movie 3, Figs. 2a and S5), while further addition of EDN3 had more variable effects, presumably caused by the exhaustion of Ca²⁺ reservoirs upon repeated stimulation with EDN3. In stage E12.5

ileum, CA also increased threefold at 1 nM EDN3 (Fig. S5), and the application of 10 nM EDN3 induced an immediate increase of intracellular Ca²⁺ across all ENCCs (Fig. 2b, Supplementary Movie 3, Fig. S5). Unlike EDN3, GDNF, the other major ligand involved in ENCC development, did not modify CA at the 10 ng/mL concentration known to drive ENCC chemotaxis[24] (Figs. 2c and S6). EDN3 is endogenously expressed by the gut mesenchyme, concentrating at the cecum and at the anti-mesenteric ileal border[25,26]. This pattern mirrored the CA heatmaps we recorded (Figs. 1a and S3). We therefore blocked endogenous EDN3/EDNRB signaling with the EDNRB antagonist BQ788: it almost extinguished CA ($-77 \pm 14$ %, $n = 14$) (Figs. 2d and S5, Supplementary Movie 4) for at least 24 h (Fig. 2e), while the drug vehicle, DMSO, did not affect CA (Fig. S7). In addition, BQ788 systematically led to a rounding of ENCCs and a retraction of their cell processes (Fig. 2f, Supplementary Movie 4). We finally tested whether CA differed in an EDN3 missense mutation model that develops Hirschsprung disease, the ls/ls mouse. Because this mouse didn't express GCaMP, we resorted to 1 day on-substrate-culture of intestinal explants in medium with GDNF and loaded the preparation with Fluo4-AM. The average frequency of ENCC Ca²⁺ transients was lower by ~25% in the ls/ls homozygote than in controls (Fig. 2g-i).

### T-type Ca²⁺ channels and Cl⁻ channels mediate Ca²⁺ oscillations

We next investigated the ionic transport mechanisms leading to Ca²⁺ oscillations downstream of EDNRB. Removing extracellular Ca²⁺ with EDTA led to a complete cessation of CA (Fig. 3a, Supplementary Movie 5). Consistent with the requirement for extracellular Ca²⁺, GdCl₃, a cation channel blocker, significantly reduced CA (Figs. 3a and S8). Interestingly, transients could still be elicited by acute administration of EDN3 10 nM in the presence of EDTA (Figs. 3e and S9, Supplementary Movie 5), indicating a contribution of intracellular Ca²⁺ stores. 2-APB halted activity (Figs. 3a and S8) while ryanodine did not affect CA (Figs. 3a and S8). These findings show that both extracellular Ca²⁺ and intracellular IP3-, but not ryanodine-sensitive Ca²⁺ stores contribute to CA.

We further questioned the entry pathway of extracellular Ca²⁺, first considering voltage-gated Ca²⁺ channels (VGCCs). The L-type Ca²⁺

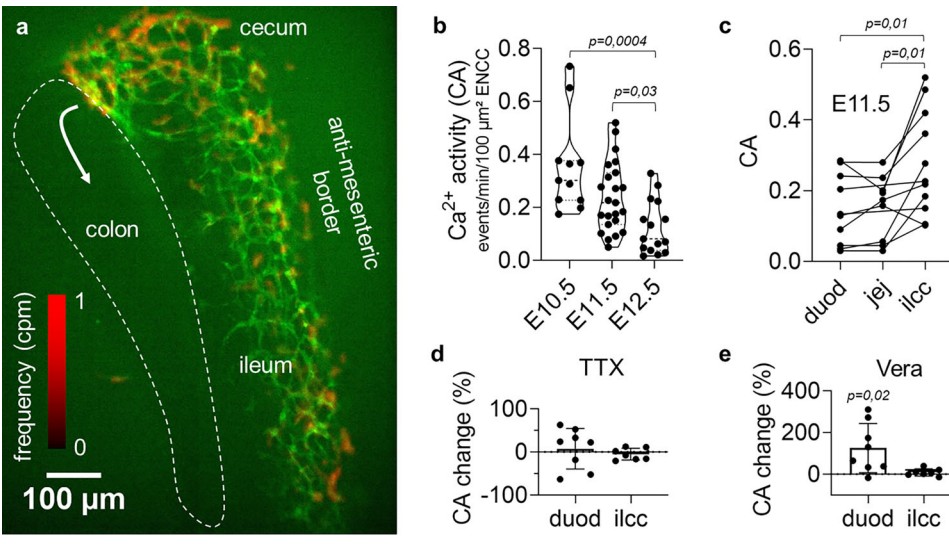

**Fig. 1 | Endogenous Ca²⁺ activity in enteric neural crest cells during gut colonization. a** Green: average time-projection of GCaMP time-lapse at the E11.5 ileo-cecal junction (Supplementary Movie 1); ENCCs have reached the cecum and are about to enter the colon (white arrow). Red: CA frequency heatmap shows concentration of activity at the cecum and the ileum anti-mesenteric border (see also Fig. S3). **b** CA at the ENCC migration front at stages E10.5 (jejunum, $n = 11$), E11.5 (ileo-cecal junction, $n = 23$), E12.5 (colon, $n = 15$), Kruskall Wallis test. **c** CA in the

duodenum (duod), jejunum (jej) and ileo-cecal junction (ilcc) at E11.5. Lines link points measured from the same sample ($n = 11$), two-tailed Wilcoxon matched pairs signed rank (WMP) test. **d** Tetrodotoxin (TTX, 1 µM) did not modify CA in the duodenum ($n = 8$) or ilcc ($n = 7$) at E11.5. **e** Veratridine (Vera, 10 µM) induced CA in the duodenum ($n = 8$), but not in the ilcc ($n = 7$) at E11.5, WMP test. All error bars are ±SD.

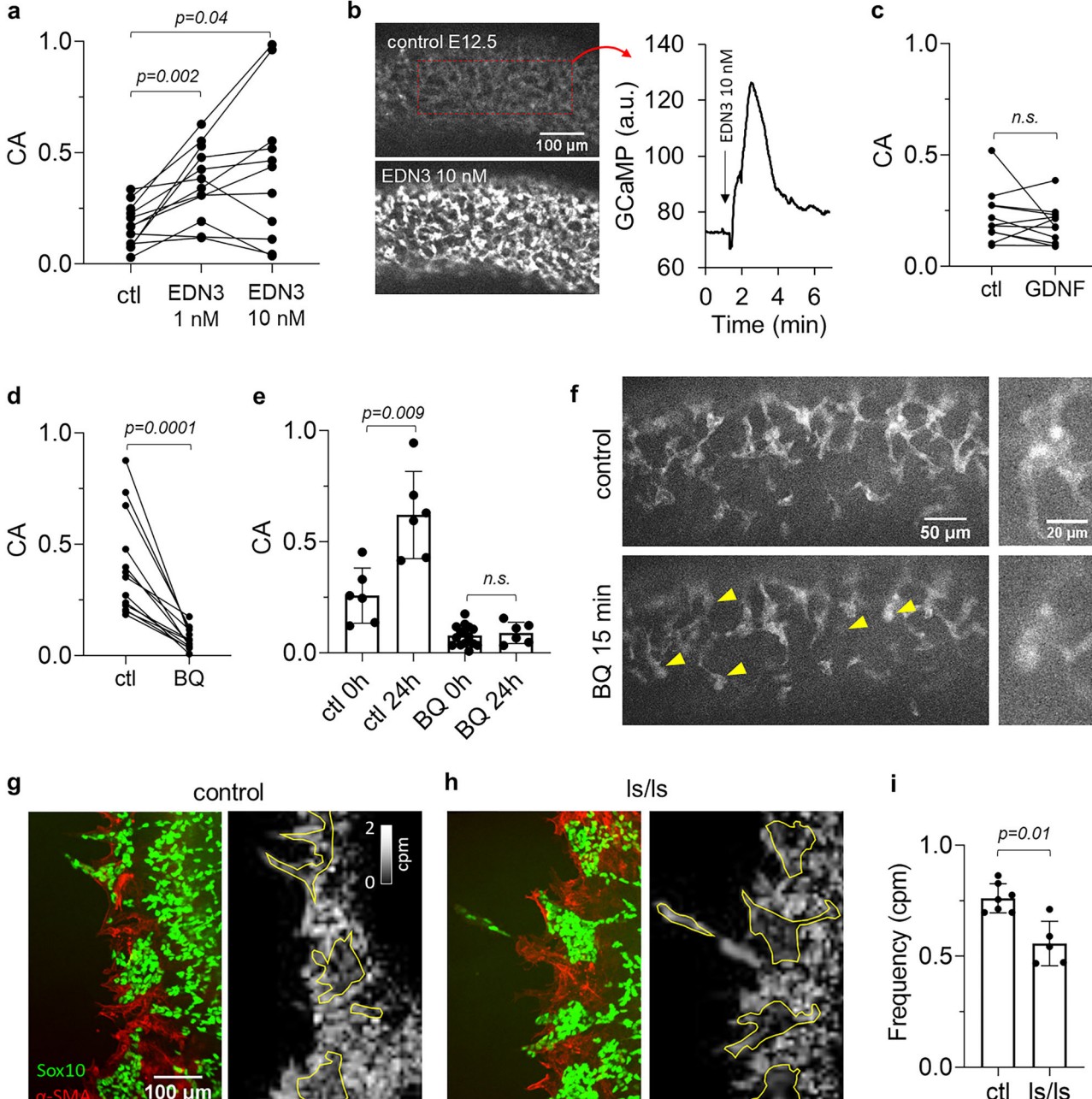

**Fig. 2 | EDN3/EDNRB drives Ca²⁺ activity in ENCCs. a** Effect of EDN3 1 and 10 nM on Ca²⁺ activity (CA) in E11.5 ilcc, $n = 12$, WMP test. **b** Sharp Ca²⁺ rise spanning all GCaMP positive cells after 10 nM EDN3 administration detected in 11/13 samples in E12.5 ileum (Supplementary Movie 3). Right panel: GCaMP signal intensity in the dashed red rectangle upon EDN3 introduction. The intracellular Ca²⁺ rise could not be stimulated again upon renewed application of 10 nM EDN3 ($n = 6/6$). **c** Effect of GDNF 10 ng/mL in E11.5 ilcc, $n = 11$, WMP test. **d** Effect of the EDNRB blocker BQ788 10 μM, E11.5 ilcc, $n = 14$, WMP test. **e** Evolution of CA in control ($n = 6$) and BQ 788 ($n = 6$) treated samples after 24 h in culture, two-tailed Mann Whitney test. Error bars are ±SD. CA increase of control samples after 24 h culture does not reflect the physiological, stage-by-stage decrease of CA (Fig. 1b): this discrepancy could arise

from components of the medium, possibly serum. **f** Left: ENCC process retraction and cell rounding (yellow arrowheads) observed after 15 min incubation in BQ 788 (Supplementary Movie 4). Right: magnified appearance of a cell group before/after BQ 788 application. **g, h** Left panels: Sox10 and α-SMA immunohistochemistry of ENCCs and mesenchymal cells in control (heterozygote) and ls/ls sample. Right panels: registered heatmap of Ca²⁺ transient frequency, ENCC aggregate borders are shown in yellow as drawn from the IHC images. ENCCs display lower frequency than the mesenchyme. **i** Frequency of transients for $n = 7$ controls and $n = 5$ ls/ls, two-tailed Mann Whitney test. Each dot is a different sample/embryo and a line connects the same sample before and after drug application in all figures of this report. Error bars are ±SD.

---

channel (CaV1.2) blocker nifedipine inhibits activity in fetal and adult ENS neurons[27] but did not significantly alter CA in E11.5 guts (Figs. 3b and S10). Ruthenium red, a non-selective blocker of P/Q-type (CaV2.1) and N-type (CaV2.2) channels as well as of cationic TRPV channels, did not affect CA either (Figs. 3b and S10). Blockade of all 3 T-type Ca²⁺ channels (CaV3.1,2,3) with Z944 induced a dose- and time-

dependent decrease of CA (Figs. 3b, S10, and S12, and Supplementary Movie 6). CaV3.2 (encoded by *Cacna1h*) is strongly expressed in E11.5 ENCCs[28]; we found that specific blockade of CaV3.2 with ascorbic acid (AA) 1 mM[29] also induced a dose- and time-dependent decrease of CA (Figs. 3b, S10, and S12, and Supplementary Movie 6). The only commercially available T-type Ca²⁺ channel agonist, SAK3, is specific for

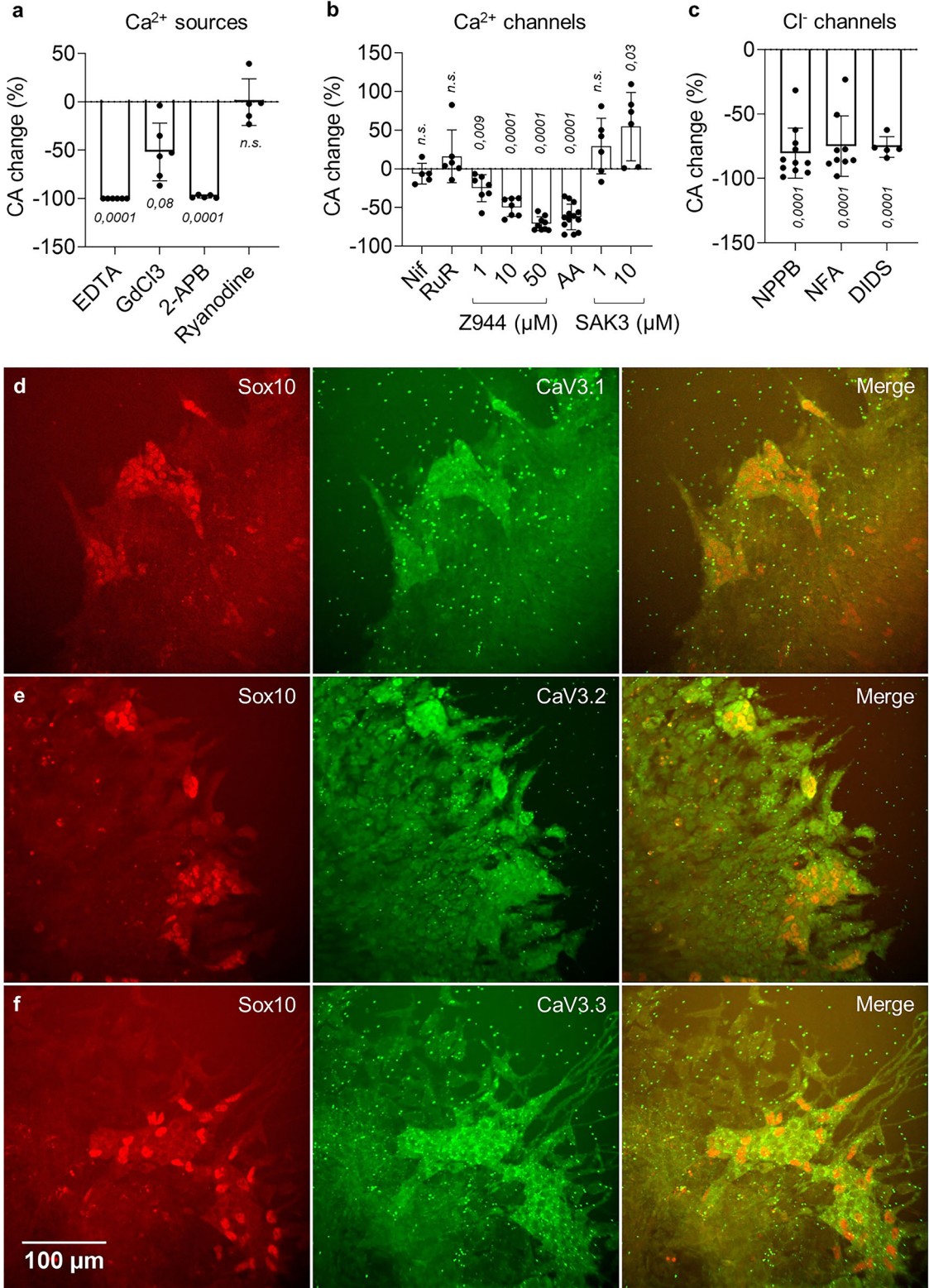

**Fig. 3 | T-type Ca²⁺ channels and Cl- channels are critical for Ca²⁺ oscillations.** All experiments in (**a**–**c**) are performed at E11.5 at the ileo-cecal junction. Error bars are ±SD. **a** CA change relative to control before drug application for EDTA (2 mM, $n = 6$), GdCl3 (100 μM, $n = 6$), 2-APB (100 μM, $n = 5$), ryanodine (10 μM, $n = 5$). **b** CA change for nifedipine (Nif, 10 μM, $n = 5$), ruthenium red (RuR, 100 μM, $n = 6$), Z944 at 1 μM ($n = 7$), 10 μM ($n = 7$), 50 μM ($n = 9$), ascorbic acid (AA, 1 mM, $n = 13$) and SAK3 at 1 μM ($n = 6$), 10 μM ($n = 6$). **c** CA change for NPPB (100 μM, $n = 11$), NFA (50 μM,

$n = 9$) and DIDS (500 μM, $n = 5$). Two-tailed $p$-values for (a-c) are calculated from the Student t-test. Sox10 and CaV3.1 (**d**), CaV3.2 (**e**) and CaV3.3 (**f**) IHCs of ENCCs migrating from an intestinal explant on a substrate. All three T-type Ca²⁺ channels are expressed by ENCCs, but not by the surrounding mesenchymal cells. The small bright green dots are AF647 μ-beads fluorescing in the same range as the T-type Ca²⁺ channel secondary antibody. They were added in the overlying collagen gel for another experiment (see Fig. 6).

CaV3.1 and CaV3.3[30]. This molecule induced a significant CA increase at 10 μM (Figs. 3b and S10, and Supplementary Movie 6), suggesting that all three T-type $Ca^{2+}$ channels are involved in ENCC CA. IHC for CaV3.1, CaV3.2 and CaV3.3 (Fig. 3d-f) showed that all three channel types were indeed expressed by ENCCs (Sox10+ cells), but not by the surrounding mesenchyme. SAK3 also led to a significant CA increase post-EDNRB blockade with BQ788, but CA remained well below physiological levels, even after 1 day culture (Fig. S14).

We next targeted $Cl^-$ channels by applying three different classes of inhibitors: 5-Nitro-2-(3-phenylpropylamino)benzoic acid (NPPB), niflumic acid (NFA) and 4,4′-Diisothiocyanato-2,2′-stilbenedisulfonic acid (DIDS). These compounds led to a drastic CA decrease (Figs. 3c and S11, and Supplementary Movie 7) and a retraction of cell processes (Supplementary Movie 7). Transient CA could also be elicited by EDN3 10 μM after 15 min incubation in 100 μM NPPB, but the transients had low amplitude (Fig. S9). NPPB and DIDS systematically induced ENCC death after 24 h (Fig. S15). NFA 50 μM inhibition kinetics were similar to Z944 50 μM (Fig. S12): CA had recovered after 24 h, ENS morphology was normal, and the fraction of dead ENCCs similar to control samples (Fig. S16). $Cl^-$ channels likely contribute to $Ca^{2+}$ transients by depolarizing the cell membrane through $Cl^-$ efflux, thereby opening VGCCs[31,32]. This idea is in line with the fact that the membrane depolarizer 4-AP markedly increased CA (Fig. S13). 4-AP could not however re-stimulate CA post-EDNRB block (Fig. S14c), and also induced cell death after 24 h (Fig. S15).

The purinergic pathway is an important actor of ENCC multi-cellular $Ca^{2+}$ waves[23]. We found that the $ATP_e$ enzymatic degradation inhibitor ARL 67156 and exogenous $ATP_e$ could transiently increase wave activity after EDNRB blockade (Fig. S14d,e). Higher $ATP_e$ concentrations could even result in a network-spanning intracellular $Ca^{2+}$ rise (Fig. S14d and Supplementary Movie 8). These effects were however not sustained in time (Fig. S14e), indicating that the purinergic pathway could not bypass the main regulator EDN3/EDNRB.

### $Ca^{2+}$ activity correlates with ENCC migration speed

We next investigated how pharmacological upregulation of CA by EDN3 or SAK3 affected ENCC migration from E11.5 midgut explants embedded in a 3D collagen gel. We found that both EDN3 1 nM and GDNF 10 ng/mL increased migration distances from the explant compared to control conditions, and that they acted synergistically when combined (Fig. 4). Strikingly, we found that the CA activator SAK3 was able to mimic the effect of EDN3, i.e. migration distances were similarly high in GDNF + SAK3 10 μM and GDNF + EDN3 1 nM conditions (Fig. 4).

We next assessed the effect of CA inhibitors on ENCC migration in the colon in full E11.5 guts cultured for 1 day in agarose (a gel not permissive to migration). We found that the four conditions that significantly and long-lastingly (Figs. S12 and 2e) lowered CA - Z944 50 μM, NFA 50 μM, AA 1 mM and BQ788 10 μM - all decreased ENCC migration distances in the colon (Fig. 5a,b). These effects could not be attributed to toxicity, as cell death in most ($n = 49/54$) samples was on the same level as control samples (Fig. S16). Addition of SAK3 to BQ788 did not improve ENCC migration down the colon, consistent with the fact that CA was still very low in these conditions (Fig. S14). These results show that, in addition to the well-known inhibitory effect of BQ788 on ENCC migration[33,34], $Cl^-$ channel and T-type VGCC inhibition also result in a migration defect leading to colonic aganglionosis.

### $Ca^{2+}$ activity drives cell contractility and traction force to the extracellular matrix

ENCCs exert a traction force on the tissue as they migrate[24], visible in-situ as a backflow of mesenchyme (Supplementary Movie 9). Several observations indicated that $Ca^{2+}$ oscillations could modulate this traction force by influencing cell contractility, as in smooth muscle: 1°) CA blockers led to process retraction and rounding (Fig. 2f,

Supplementary Movie 4,7), indicating a loss of ENCC-extracellular matrix (ECM) traction; 2°) In a 2D migration assay at x60 magnification, we noticed that $Ca^{2+}$ transient could induce abrupt lamellipodium detachment from the substrate ($n = 4$ such events observed, Fig. S17a, Supplementary Movie 10). This could be driven by actomyosin or by calpain activity[35]; 3°) Most $Ca^{2+}$ transients (79/102) in this assay were correlated with an abrupt change of nucleus speed, as measured by deep-learning assisted tracking and by kymographs (Fig. S17b-c, Supplementary Movie 11).

To quantify the force exerted by ENCCs on the ECM, we let ENCCs migrate from E11.5 midgut explants in a collagen gel seeded with fluorescent beads that served as fiducial markers of gel deformation (Fig. 6a inset). The traction force on the collagen scaffold induced bead displacements towards the migration front (Fig. 6, Supplementary Movie 12). Immediately after induction of CA by addition of EDN3 1 nM, the bead speed abruptly increased (Fig. 6, Supplementary Movie 12), and the tip of the migration "fingers" accelerated (Supplementary Movie 12). When EDNRB was blocked by BQ788, the beads suddenly relaxed, indicating a sudden loss of ENCC-ECM traction force (Fig. 6a,b,d, Supplementary Movie 12), and the migration fingers retracted (Supplementary Movie 12). The effect of the T-type $Ca^{2+}$ channel blocker Z944 was similar to that of BQ788, although the decrease in traction force was less pronounced (Fig. 6c,e). This experiment shows that, similarly to endothelin-induced smooth muscle contraction, EDN3 could stimulate $Ca^{2+}$-dependent actomyosin, increasing the ENCC-ECM traction force necessary for migration, whereas blocking EDNRB or T-type $Ca^{2+}$ channels relaxed it.

## Discussion

Endothelins were identified in 1987 as potent modulators of vascular smooth muscle (vSMC) contractility[36]. When binding to EDNRs on vSMCs, endothelins drive an increase in cytosolic $Ca^{2+}$ mediated both by the release of $Ca^{2+}$ from intracellular stores and by the entry of extracellular $Ca^{2+}$ via CaV1.2 VGCCs[32,37,38]. The opening of CaV1.2 is favored by the efflux of $Cl^-$ from the cytosol to the extracellular medium, resulting in membrane depolarization[31,32]. Increased cytosolic $Ca^{2+}$ leads to calmodulin activation and acto-myosin cross-bridges responsible for vSMC contraction. As the roles of endothelins in blood pressure regulation became more prominent, mutant mice for EDN3[39,40] and EDNRB[41] mouse were re-examined. The phenotype of these mice came as a surprise: they presented with colonic aganglionosis at birth, a phenotype known as Hirschsprung disease (HD). EDN3/EDNRB has later been identified as a key pathway promoting ENCC proliferation and delaying their differentiation to enteric neurons and glia[10,42,43]. The EDN3/EDNRB pathway has since then been attributed a dual, "moonlighting" role[14,44]: regulator of vSMC tone and blood pressure in adult physiology, driver of ENCC proliferation in the embryo. We reveal here that the immediate action of EDN3 on ENCCs is in fact nearly identical to its effect on smooth muscle tone: it triggers $Ca^{2+}$ activity (Figs. 1 and 2), via a very similar molecular route (Figs. 3 and 7) to vSMC, and the increased cytosolic $Ca^{2+}$ oscillations enhance migration (Figs. 4 and 5) by inducing cell contractility (Fig. 6). vSMC contraction leads to vessel constriction; ENCC contraction leads to an increased traction force to the extracellular-matrix, that is necessary for their migration inside the gut mesenchyme. This force is the reason why investigators have found it indispensable to pin[45] or embed the gut tract during ex-vivo migration assays[46], as it otherwise leads to tissue shrinkage and improper migration. The traction force is transmitted via β1-integrins (Fig. 7), and ENCC-specific β1-integrin mutants have been shown to present with a HD phenotype[47–49]. $Ca^{2+}$ signaling defects reduce the traction force of the ENCCs to the extracellular matrix, lowering their migration speed and thus resulting in colonic aganglionosis. Our investigation shows that ENCCs are akin to miniature muscles that contract and crawl in response to a constrictor peptide, endothelin 3.

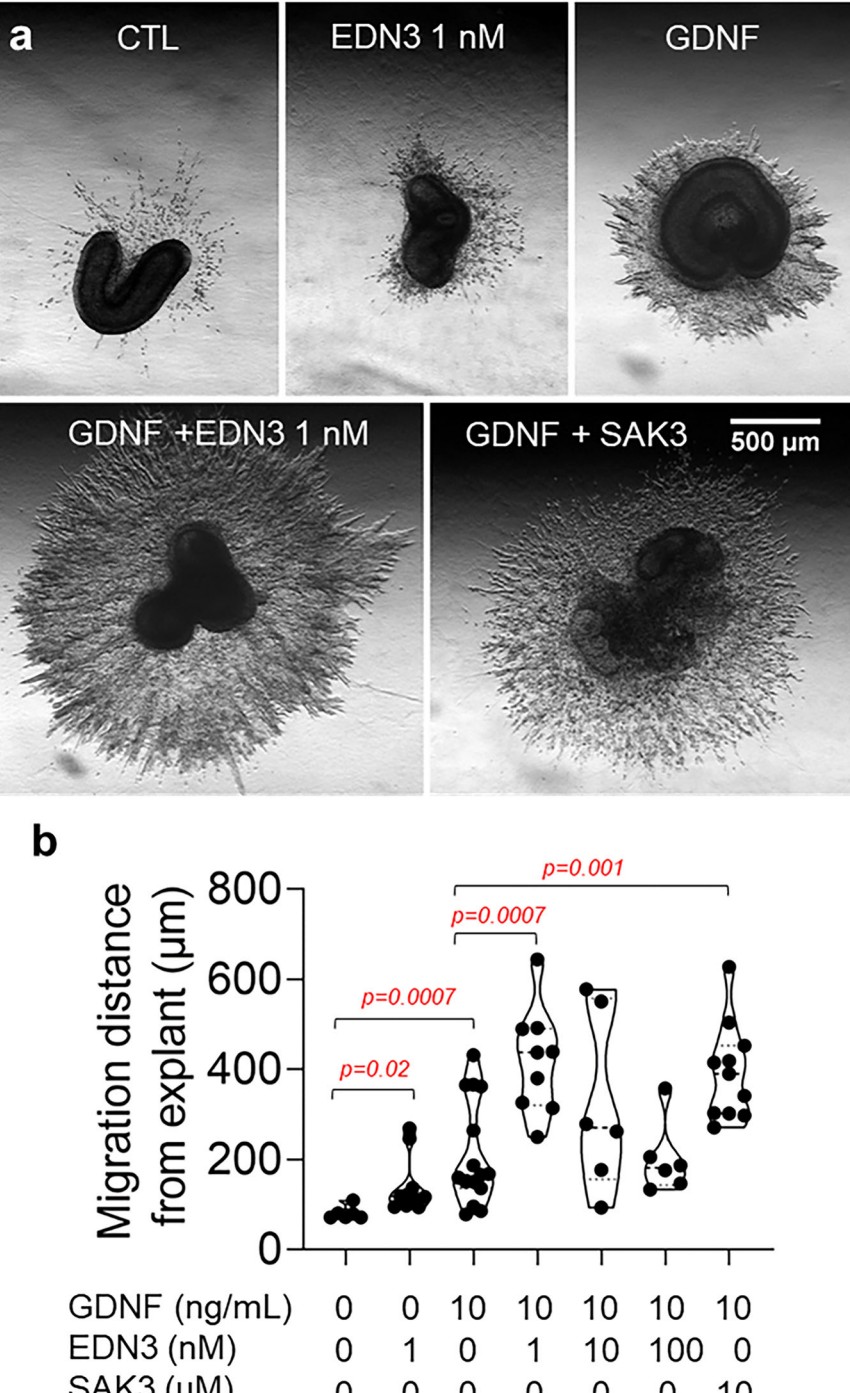

**Fig. 4 | Ca²⁺ activity enhancers EDN3 and SAK3 promote migration in a collagen gel migration assay. a** E11.5 midgut explants embedded in a collagen matrix and cultured for 3 days in control, EDN3 1 nM, GDNF 10 ng/mL, GDNF + EDN3 1 nM and GDNF + SAK3 10 μM conditions. ENCC migration in the collagen is visible in brightfield Schlieren-type lighting as a halo of radially oriented cells surrounding the darker explant. **b** Average migration distance from the explant after 3 days in the different drug conditions, two-tailed Mann-Whitney test. Samples numbers in each group, from left to right, are $n = 6, 10, 15, 9, 6, 6, 11$. Violin plots present the median, upper and lower interquartile range.

It is likely that cytosolic Ca²⁺ oscillations are also involved in the long-term effects of EDN3/EDNRB on ENCC proliferation. ENCC proliferation and migration are inexorably linked – ENCC cannot invade the colon or a collagen gel if they do not also proliferate[50]. Here, we reported macroscopic migration distances (linear in the colon, spherical in the gel) rather than local proliferation rates. EDN3/EDNRB also delays ENCC differentiation[10,42,43]. Higher CA is generally associated with an undifferentiated state in stem cell cultures[16,17]. The mitogen activated protein kinase (MAPK) and the phosphoinositide 3-kinase (PI3K) pathways can both be induced downstream of EDNRB[51], and are dependent on Ca²⁺ activity[52,53] (Fig. 7). MAPK family members extracellular signal-regulated kinase (ERK) and c-Jun N-terminal kinase (JNK) both promote ENCC migration[54], and likely also regulate ENCC proliferation and differentiation.

T-type receptor blockade by mibefradil 1 μM was previously reported not to affect ENCC migration[28]. It is probable that this

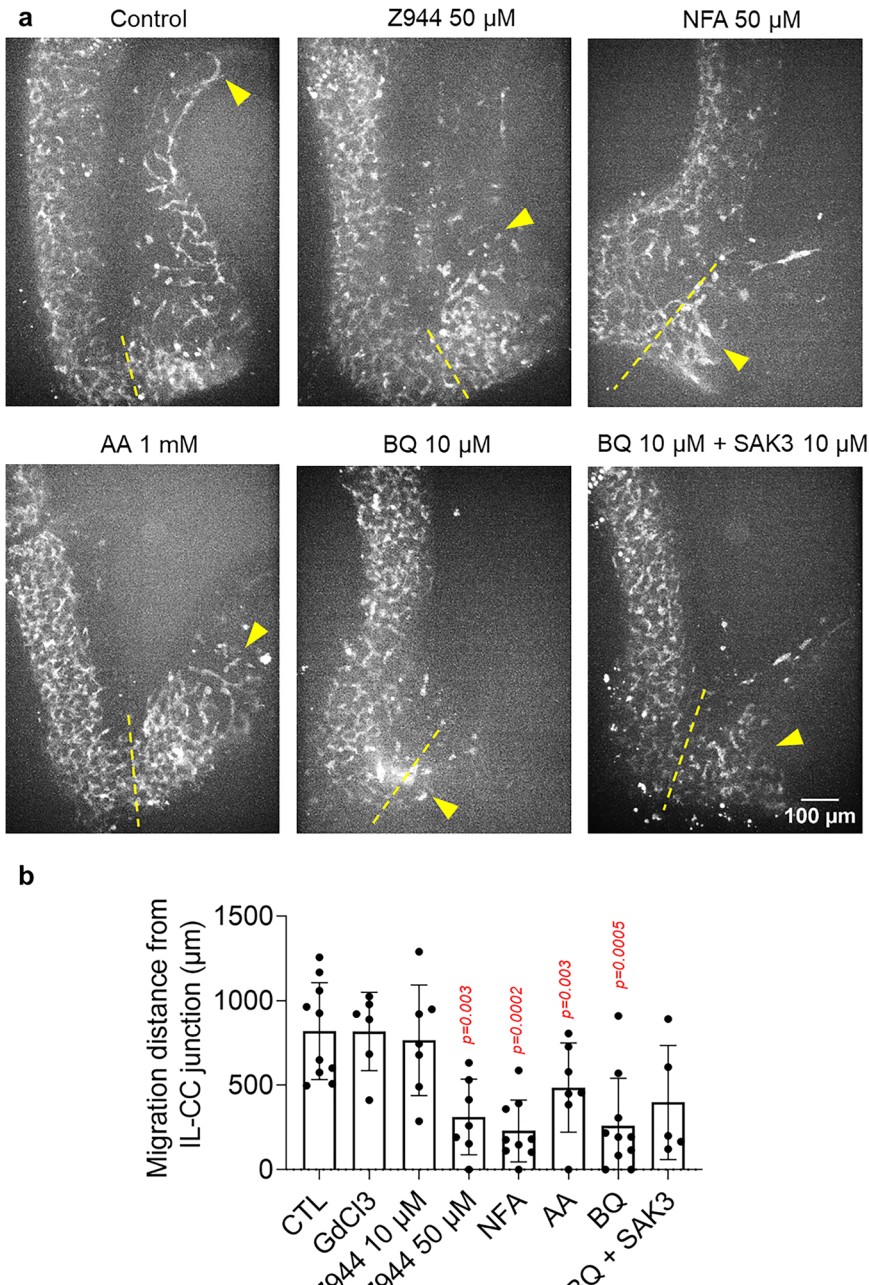

**Fig. 5 | Inhibition of Ca²⁺ activity slows down ENCC migration in the cecum and colon. a** Maximum projection of GCaMP z-stacks of E11.5 guts cultured for 1 day in control conditions (with DMSO vehicle alone) and in various chemical conditions that alter CA. The dashed yellow line marks the position of the the ilcc junction, and the yellow arrowhead the position of the migration front after 1 day culture. AA= ascorbic acid. **b** Migration distance in the colon from the ilcc junction for control ($n = 10$), GdCl3 100 µM ($n = 6$), Z944 10 µM ($n = 7$), Z944 50 µM ($n = 7$), NFA 50 µM ($n = 9$), AA 1 mM ($n = 7$), BQ788 10 µM ($n = 10$), BQ788 10 µM + SAK3 10 µM ($n = 5$). Two-tailed $p$-values are shown for the Mann-Whitney test compared to the control group. Z944 50 µM, NFA 50 µM, AA 1 mM and BQ788 10 µM significantly slowed down migration. BQ + SAK3 did not improve migration compared to BQ alone. Error bars are ±SD.

concentration was insufficient to cause aganglionosis, because the IC50 of mibefradil on isolated cells is 3 µM[55]; in our experience, effective, sustained blocking of channel activity in whole-gut cultures requires concentrations far above drug IC50 to yield a migration phenotype. Hirst et al.[28]. also blocked Cl⁻ channels with NPPB 100 µM from an only 100-fold stock solution in a non-specified solvent, and found no effect on migration or on ENCC survival. In our experiments, NPPB 100 µM disrupted CA and migration by systematically inducing ENCC death ($n = 11/11$, Fig. S15). The choice of NPPB solvent or inconsistencies in the actual concentrations applied by Hirst et al. may explain the discrepancy. More generally, we were guided in our choices of pharmacological compounds and concentrations by the Ca²⁺ response, whereas Hirst et al. applied these compounds "blindly" to wildtype guts. Our study differs methodologically from the pioneering work of Hao et al.[23]. These investigators focused on multicellular Ca²⁺ waves, with frequencies of ~10⁻⁴ waves/min/100 µm². This is consistent with the wave activity we measure, but ~100-1000 time more frequent events occur as single-cell transients and are the main form of ENCC CA. We found that, although ATP$_e$ and ARL 67156 can indeed promote multicellular waves[23], they only play a modulatory role in a mechanism that is dominated by EDNRB, Cl⁻ and CaV3 channels.

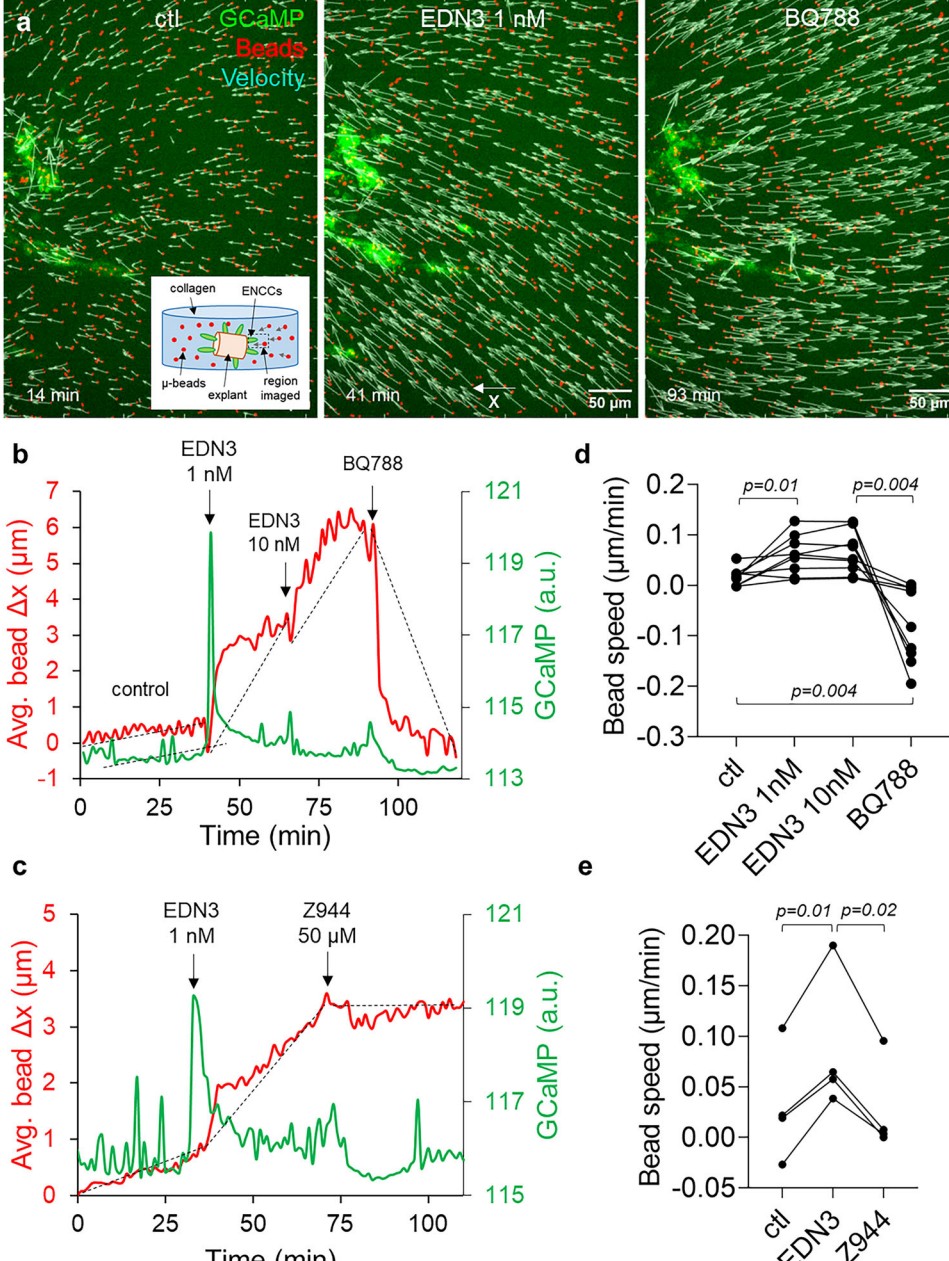

**Fig. 6 | Ca²⁺ transients promote migration by increasing ENCC traction force to the extracellular matrix. a** ENCC 3D migration from a E11.5 midgut explant (to the left, not in the field of view), after 24 h in a collagen matrix with 10 ng/mL GDNF and fiducial marker beads. Maximum projections of z-stacks shows ENCCs to the left (green, GCaMP signal), and AF647 beads (red) inserted in the collagen matrix. The velocity vectors (cyan) are deduced from bead tracking in 1 stack-per-minute videos (Supplementary Movie 12). Left panel: initially, the beads move toward the ENCCs because of the traction force they exert on the collagen gel. The inset shows the experimental setup. Middle panel: the traction force sharply increases when Ca²⁺ rises after EDN3 1 nM administration. Right panel: the traction force inverts (relaxes) after application of BQ788. **b** Average bead x displacement from their initial x position versus time (red curve), and average GCaMP signal measured in an ROI surrounding the ENCCs (green), in a control, EDN3 1 nM, EDN3 10 nM and BQ788 10 μM sequence and (**c**) a control, EDN3 1 nM and Z944 50 μM sequence. The average bead displacement was performed over all beads that were not in direct contact with the ENCCs. The average bead speed for each condition is defined as the slope of the dashed lines. Note that the time-resolution of this experiment (1 z-stack per minute) did not allow to resolve in detail the Ca²⁺ transients, but only occasional Ca²⁺ bursts (edn3) or the absence of activity (BQ788, Z944). **d** Average bead speed for $n = 9$ control - EDN3 1 nM – EDN3 10 nM – BQ788 10 μM experiments and (**e**) $n = 4$ control - EDN3 1 nM – Z944 50 μM experiments, two-tailed paired Student t-test.

CaV3.1 and CaV3.2 expression in E11.5 ENCCs has been previously measured by PCR[28]; CaV3.2 was found to be ~640 times more expressed than CaV3.1. Our IHC, CA and migration assays indicate that all three CaV3 channel types are present and functional in ENCCs. The CaV3.1 & CaV3.3 agonist SAK3 promoted CA (Fig. 3) and ENCC migration (Fig. 4); the CaV3.2 antagonist ascorbic acid reduced CA (Fig. 3) and ENCC migration (Fig. 5). Very interestingly, a single-nucleotide polymorphism (SNP) of the *CACNA1H* gene encoding CaV3.2 has been uncovered in a recent HD exome-wide association study[15]. Although it is not known whether this point mutation altered CaV3.2 expression or function, our findings suggest it induces HD by altering Ca²⁺ signaling. This motivates further research using dedicated mouse models to better understand the causes of neurocristopathies. We note that the CaV3.2 -/- mouse is viable[56,57], although the mutation

induces more pre-natal lethality[58]. It is possible that the surviving CaV3.2 -/- embryos develop compensatory $Ca^{2+}$ influx mechanisms, a common behavior when only one of several protein isoforms is knocked-out[59].

We found that GDNF and EDN3 (or SAK3) have a synergistic action on ENCC migration from explants (Fig. 3). Our results are in agreement with the findings of Bergeron et al. [60]. We did not observe that EDN3 and GDNF were antagonistic, as has been reported in other studies with mouse[25], rat[61] and chicken embryos[46]. These studies were performed respectively at stages E10.5-E11, E13 and E8, at EDN3 concentrations of 100, 20 and 100 nM and after 16 h, 2-3 days and 3 days of migration. It is difficult to fathom the reasons of this discrepancy but we stress that: 1°) migration after 16h[25] is too scarce to be precisely quantified, 2°) the EDN3 concentrations applied by other investigators are 1-2 orders of magnitude higher than the maximum pro-CA and pro-migratory effect we report at 1 nM. 1 nM likely reflects the concentration encountered physiologically by ENCCs as they migrate down the gut mesenchyme. We found that CA was not significantly increased at 10 nM EDN3 compared to 1 nM (Figs. 2a and S5c), indicating a saturation effect. Administration of EDN3 10 nM at E12.5 gave rise to a single pan-ENCC transient, which, although impressive, most likely never occurs physiologically. The pan-ENCC $Ca^{2+}$ surge induced by exogenous EDN3 may occur only at E12.5 (and not at E11.5) because of a reduced saturation of EDNRB receptors by endogenous EDN3 at this stage, making it more sensitive to exogenous application.

Edn3 expression is highest in the cecum, with little expression in the hindgut at E12[26], which correlates with reduced CA at the ENCC wavefront at E12.5 (Fig. 1). These observations suggest that EDN3 may be most necessary up until the cecum is traversed; slowing down of ENCCS by CA activity inhibitors in our ex-vivo assay (Fig. 5) may have primarily affected migration through the cecum, resulting in a paucity of ENCCs in the colon. Colonic aganglionosis can result from slowed-down ENCC migration at any point of their journey down the gut, not necessarily from slower migration in the colon.

We found that ls/ls ENCCs only displayed a ~ 25% reduction in CA compared to controls. This experiment was performed in the presence of GDNF, serum, and after 1 day culture, all of which can artificially hike CA (Fig. 2e) compared to its in-vivo state. A more refined approach based on GCaMP expressing ls/ls mutants would, given our results on the effect of EDNRB blockade, likely reveal a much more drastic CA deficit. Testing the hypothesis that $Cl^-$ channels drive an efflux of $Cl^-$ ions and a concomitant depolarization (Fig. 5) as in smooth muscle[31,32] will require electrophysiological measurements. The identification of the $Cl^-$ channels involved hinges on the future development of more specific antagonists and agonists: ENCCs express many different $Cl^-$ channels[28] and several types may be involved. We have not found a correlation between mutations affecting $Cl^-$ channels and HD pathogenesis in the literature. This of course does not preempt the fact that they are important for ENCC migration, as such mutations may plainly be lethal. Although our pharmacological approach allowed us to specify CaV3 channels as an important gateway of extracellular $Ca^{2+}$, we cannot at this stage exclude other non-VGCC entry mechanisms, like TRPC[62].

We deciphered an important mechanism by which endothelin 3 triggers $Ca^{2+}$ activity and contractility

necessary for the migration of enteric neural crest cells. This mechanism is likely to play a more general role in NCC derived melanocytes or Schwann cells and EDNRA-expressing cranial NCCs[14], potentially linking $Ca^{2+}$ activity to other neurocristopathy syndromes[3]. Melanoma[7] and neuroblastoma[63] are known to recapitulate many NCC traits. T-type $Ca^{2+}$ channel upregulation is associated with melanoma aggressiveness[64] while $Ca^{2+}$ signaling is altered in neuroblastoma[65]. The endothelin axis is more generally aberrantly expressed in tumors[66],

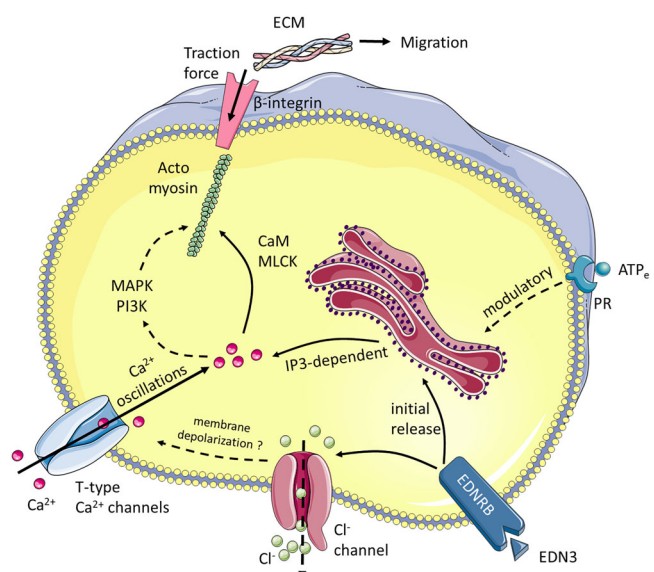

**Fig. 7 | Synthetic scheme of the pathways leading to $Ca^{2+}$ transient generation and ENCC migration.** PR: purinergic receptor, IP3: inositol trisphosphate, CaM: calmodulin, MLCK: myosin light chain kinase, MAPK: mitogen activated protein kinase, PI3K: phosphoinositide 3-kinase. Dashed arrows indicate mechanistic links for which there is evidence from the literature in related systems. EDNRB activation can trigger a direct release of $Ca^{2+}$ from intercellular stores, but $Cl^-$ channels, T-type $Ca^{2+}$ channels and extracellular $Ca^{2+}$ are necessary for sustained $Ca^{2+}$ oscillations. Image elements provided by Servier Medical Art (https://smart.servier.com), licensed under CC BY 4.0 (https://creativecommons.org/licenses/by/4.0/).

promoting invasion and metastasis: the mechanism we describe here for neural crest cells may also be at play in cancer cell migration.

## Methods

### Ethics
Mice were hosted at the Institut Jacques Monod (GCaMP) and at the Institut Imagine LEAT (ls/+ heterozygotes) animal husbandries. They had access to housing, food and water ad-libitum. Temperature in the husbandry was 20–24 °C and humidity in the range 45–55%. The mice were exposed to light from 7 am to 7 pm. Females were kept in common cages with a maximum of 5 females per cage. For mating, the male is introduced in the evening and removed in the morning after detection of plugs in the morning. Pregnant mice were killed by cervical dislocation to retrieve embryos age E10.5 to E12.5. The embryos were separated and immediately beheaded with surgical scissors. The methods used to kill the mice conform to the guidelines of CNRS and INSERM animal welfare committees. Killing of mice for retrieval of embryos is a terminal procedure for which neither CNRS or INSERM assign ethics approval codes hence none are given here.

### Mouse gut samples
The Cre reporter mice C57BL/6N-Gt(ROSA)26Sor^tm1(CAG-GCaMP6f) Khakh/J mice is referred to as Gcamp6fl/fl. A transgenic mouse line in which the transgene is under the control of the 3-kb fragment of the human tissue plasminogen activator (Ht-PA) promoter Tg(PLATcre) 116Sdu16 is referred to as Ht-PA::Cre. GCamp6fl/fl males were crossed with Ht-PA::Cre females to generate embryos carrying the calcium fluorescent reporter in neural crest cells and their derivatives[27]. The gut of each embryo was dissected in PBS with 1 mM $Ca^{2+}$, 0.5 mM $Mg^{2+}$ and 1% penicillin-streptomycin, from stomach (cut at the oesophagus-stomach junction) to colon (cut at the colon-anus junction). The mesentery of E11.5 guts was kept intact. 57 % (118/207 for which the full count was performed) of the embryos expressed GCaMP specifically in

NCCs, while 43 % expressed GCaMP in all cells, with the mesenchymal signal being dominant[67]; these phenotypes could be easily recognized during confocal imaging. We do not know the reason of this partial specificity of GCaMP. Only embryos expressing GCaMP in NCCs were used for $Ca^{2+}$ imaging. Ubiquitously GCaMP expressing embryos, which were otherwise morphologically normal, were used for experiments where $Ca^{2+}$ imaging was not required.

Heterozygous ls/+ animals were crossed and embryos retrieved at E11.5. DNA extraction and subsequent genotyping were performed from head biopsies of each embryo, using the direct PCR lysis reagent (Viagen). Genotyping was performed using primers (Eurogentec, Belgium) m-Edn3 ls F GAC TGT GCC CTA TGG ACT GT and m-Edn3 ls R GGT GAC ATC TCT GGT GCG TG[68,69].

### Organ culture and $Ca^{2+}$ imaging

After dissection, each gut was placed on a pre-solidified 0.5 mL layer of 1% low-melting point agarose (Condalab 8050.11, dissolved in PBS with $Ca^{2+}$ 1 mM and $Mg^{2+}$ 0.5 mM), in a 35 mm diameter Petri dish (Greiner). This gel layer prevents adhesion of the gut and subsequent ENCC migration on the dish bottom, confining it to the organ. The gut was then immobilized by pouring additional 0.5 mL of liquid agarose on top and letting it gel at 4 °C for 10 minutes. Capillary forces of the shallow liquid agarose layer gently press down the gut, most of the time positioning the stomach, midgut and colon in a horizontal plane that was optimal for microscopy. Immobilization is crucial both for $Ca^{2+}$ imaging and to prevent shrinking / retraction of the sample during culture. The 1 mL gut & gel were covered by 2 mL of complemented DMEM:F12 Glutamax (Gibco 31331-028). Final concentrations after diffusion and dilution of the medium in the PBS-based agarose were $Ca^{2+}$ 1 mM, $Mg^{2+}$ 0.5 mM, penicillin-streptomycin 1%, Fetal Veal Serum 6.6 %, while all other molecules that are present in DMEM:F12 but not in PBS (e.g. glucose, amino acids etc.) had their concentration divided by a factor 2/3 compared to DMEM:F12 alone. Samples were incubated at 37 °C in a 5% $CO_2$ 95% air atmosphere for at least 45 min before imaging.

Calcium imaging was performed on an inverted spinning-disk microscope (Olympus IX-81, Yokogawa CSU-X1) equipped with an ILE laser-base (Andor) and a Zyla camera (Andor, resolution 1392×1040 pixel). The GCaMP signal was excited at 488 nm and the emission filtered at 497-527 nm. Time-lapse videos were recorded at x10 magnification, 400 ms exposure time, 1 Hz acquisition rate, for at least 3 min, and often longer depending on the type of experiment. We did not observe any illumination-induced bleaching or alteration of the sample. Sample basal (endogenous) $Ca^{2+}$ activity was first recorded, and the position of the ENCC wavefront determined by a z-stack at the level of the ileo-caecal junction (E11.5). Drugs were then added from stock solutions directly in the Petri dish, far from the sample, without displacing it, and homogenized by up & down movements with a 1 mL pipette. Products used in this report as well as solvent and stock concentration are listed in Fig. S18. In preliminary experiments, addition of the drug was performed during the time-lapse to capture potential immediate effects. For most drugs, CA was then recorded 10-15 min post-administration. After imaging, samples were placed back in the incubator for further culture. Time-lapse and z-stack imaging was performed after 24 h to assess ENCC wavefront position and CA. Some samples were imaged multiple times during the culture period to assess drug kinetics (Fig. S12). Some samples were fixed and processed for IHC as described below.

The experiment with ls/ls embryos required that the ENCCs migrate out of the intestine because Fluo4AM does not penetrate inside the tissue. We achieved this by placing E11.5 explants on 35 mm Greiner Petri dishes and covering them with a thin meniscus of collagen gel (0.5 mL, 1 mg/mL, as described below) to hold them down against the Petri surface. After gelling, the preparation was topped with 2 mL complemented medium with GDNF 10 ng/mL, cultured for 1 day, loaded with 1.8 μM Fluo4-AM for 5 min, and 3 min timelapse movies were recorded in 3 different locations per sample. The samples were immediately fixed after calcium imaging for IHC.

### $Ca^{2+}$ activity analysis

Analysis of CA was performed using custom-written ImageJ 1.54r macro and Matlab R2018a scripts, and is synthesized in Fig. S1. Briefly, a fixed mesh comprised of 2745 square regions-of-interest (ROIs), each 250 μm², was overlaid on the 8-bit video. A matrix of the average pixel intensity over time in each square was retrieved and input into Matlab for peak detection. The "findpeaks" function was applied, specifying, to filter out noise, a minimum peak prominence of 2 (8-bit pixel intensity unit), a minimum peak width of 3 sec, a minimum inter-peak time of 3 sec. Illumination and peak filtering settings were identical for all videos acquired, allowing quantitative comparison of GCaMP signal intensity across samples. We measured in each ROI the number of peaks, and, for ROIs in which there was at least one peak, the frequency (number/acquisition time), average duration (width at half-maxima) and average intensity ratio $I/I_0$; the data of all ROIs was collected to generate heatmaps of these variables, assess their spatial distribution, and compute their spatial average. To reflect the general activity of the region imaged, we defined the $Ca^{2+}$ activity $CA = N_{peaks}/(A_{cells}T)$, where $N_{peaks}$ is the total number of transients in a given video, $A_{cells}$ the total area of the GCaMP positive cells within the field of view and $T$ the duration of the acquisition. $A_{cells}$ was determined by first time-projecting the average intensity to obtain a crisp image of the ENCC network, and by then applying iteratively stricter Bernsen local thresholds under ImageJ, stopping just before it diverged (Fig. S1e). $A_{cells}$ should remain constant for an experiment with a fixed z and field of view. Because our procedure yielded some variability depending on $Ca^{2+}$ basal level and activity, we selected for a given experiment the biggest $A_{cells}$ and applied it to all other conditions of this experiment (e.g. before/after drug). CA was expressed in units of events/100 μm²/min; because a cell has typically an area of 100 μm², CA has values lying in the same range as frequencies expressed in cycles-per-minute (cpm).

For ls/ls samples, we followed the same procedure but used a finer mesh of 86 μm² (9890 ROIs in total) to gain resolution to differentiate ENCC from mesenchymal CA. After registration of Sox10 & α-SMA IHCs with frequency heatmaps, we drew ROIs around Sox10 positive aggregates using the ImageJ ROI manager, and then measured the average pixel intensity of these ROIs on the greyscale frequency heatmap. The average frequency was obtained by weighing the measured frequency of each aggregate by its area (i.e. bigger aggregates contributed more to the average). Area-weighted frequency and CA are proportional; we present the former in Fig. 2i because the finer mesh size used results in a higher event count compared to the standard mesh used in the remainder of the study.

### Collagen gel organ culture

Collagen gels (Figs. 4 and 6) were prepared from Cultrex Rat Collagen I (Bio-Techne) at 1 mg/mL on ice, following the manufacturer's guidelines. For 3D contractility experiments (Fig. 6), fluorescent deep-red 0.2 μm diameter spheres (Thermofisher, F8807) were added to the mixture at 1:100000 dilution to serve as fiducial tracers. Midgut segments from E11.5 embryos were cut in 2-3 pieces with micro-scissors and each segment was embedded in 1 mL of liquid collagen, that gelled after incubating at 37 °C for 45 min. 2 mL of complemented medium with 10 ng/mL GDNF were then added. Collagen gel migration was assessed after culture for 3 days at 37 °C, in a 5% $CO_2$ − 95% air humidified atmosphere. We measured the area $S_1$ occupied by the gut explant and the ENCC halo, the area $S_2$ occupied by the gut explant alone, and computed the average migration distance from the explant as $\sqrt{S_1/\pi} - \sqrt{S_2/\pi}$. This approach yielded an average radius which took into account halo asymmetries.

## Immunohistochemistry

For registration of CA with IHC (Supplementary Movie 2, Fig. 2g,h), guts were fixed in 4% PFA in PBS for 20 min, washed 3 times, then blocked and permeated in 1% BSA and 0.1% triton in PBS for 1 h, immersed in 1:200 mouse monoclonal Anti-SOX10 (Sigma, AMAB91297) for 24 h, washed 3 times, immersed in 1:500 anti-mouse AF647 secondary antibody with 1:500 βIII-tubulin FITC conjugated antibody (Abcam 224978) or 1:500 α-SMA Cy3 conjugated antibody (Sigma C6198) for 24 h, washed 3 times in PBS, and imaged in the same location as during calcium imaging. Slight translation and rotation registration corrections between the CA time lapse movie and the IHC were performed manually using the GIMP software.

For IHC of T-type $Ca^{2+}$ channels (Fig. 3d-f), we used explants from the 3D contractility assay (see below). Some of these explants touched the bottom of the Petri-dish and ENCCs can, in addition to the 3D halo, also migrate on the dish surface, as a 2D sheet, which facilitated imaging. After fixation, blocking and permeation, samples were incubated for 1 day in 1:200 mouse Anti-SOX10 and 1:100 rabbit anti CaV3.1 (Alomone Labs #ACC-021) or 1:100 rabbit anti CaV3.2 (Alomone Labs #ACC-025) or 1:100 rabbit anti CaV3.3 (Alomone Labs #ACC-009) for 24 h, washed 3 times, incubated in secondary 1:500 anti-mouse Cy3 and 1:500 anti-rabbit AF647 antibody for another day, washed and imaged. T-type $Ca^{2+}$ channel IHC was only successful when performed on fixed cells that had been induced to emigrate from an explant, without embedding or cutting; we did not observe any specific signal when native E11.5 guts were fixed, cryo-sectioned and labeled, most probably owing to degradation of the proteins when performing these steps.

## 2D migration assay

For the x60 experiments (Fig. S17), a E11.5 GCaMP midgut explant was cultured on a Petri dish with a culture surface-treated polymer coverslip bottom (Ibidi 81156), immobilized with a 500 μL meniscus of 1% low-melting point agarose gel, and cultured one day in complemented medium with 10 ng/mL GDNF. Numerous ENCCs migrated out on the coverslip and were imaged at x60 magnification at 1 Hz, for 10 to 30 min. Cell nuclei appeared as dark spots surrounded by brighter cytoplasm. They were tracked using the deep-learning Segment Anything Model 2 (SAM2) developed by Meta and implemented under QuPath[70]. Nucleus boundary and centroid tracking were very precise (Supplementary Movie 11). Nucleus centroid position was plotted together with the GCaMP signal obtained from ROI intensity measurements of the cytoplasm of the tracked cell. Kymographs were obtained with the ImageJ Reslice function performed along a rectangle encompassing the cell trajectory.

## 3D contractility assay

3D contractility experiments were performed after 1-2 day of culture in collagen gel seeded with beads (see above), at x20 magnification, acquiring z-stacks in the bulk of the gel in GFP (cells) and AF647 (beads) with a step of 2 μm over a thickness of 30-50 μm every minute for up to 120 min. Drugs were added during the time-lapse and mixed very carefully to not perturb bead positions. The z&t stacks of the cells (GFP) and beads (AF647) were first z max-projected to yield t-stacks (time-lapse). The bead t-stack was flat-field corrected (Biovoxxel toolbox under Fiji), smoothed (Gaussian blur radius 2), thresholded (Li Auto Threshold) and the initial bead positions were measured using the ImageJ Analyze Particles tool. These initial coordinated were fed into the Tracker plugin (developed by our colleague O. Cardoso) to yield the coordinates of each bead at each time point. Inconsistent tracking was filtered out by applying conditions on the maximum rate of displacement. The average displacement of beads from their initial position towards the ENCC migration front was computed.

## Statistics and Reproducibility

All sample numbers indicated in this report correspond to different embryos (guts, biological replicates). Except for ls/ls experiments which were performed on 2 litters, data for all other experiments was collected from at least 3 different litters and technically replicated at least 3 times, i.e. they were performed on 3 different days with fresh samples following the same procedure each time. Experiments on ls/ls were performed blindly, i.e., the genotype was known after data collection and analysis. In all other experiments, blinding was not possible and not relevant to this study. The sample size range is $n = 5$-23 and was determined according to the type and variability of experiment results. $p$-values reported correspond to statistical tests mentioned in the figure legends. All instances of the WMP test are two-tailed.

## Reporting summary

Further information on research design is available in the Nature Portfolio Reporting Summary linked to this article.

## Data availability

Essential data generated or analyzed during this study are included in the manuscript and supporting files. Source data are provided with this paper. Other non-essential data are available from the corresponding author upon request. Source data are provided with this paper.

## Code availability

Essential codes for calcium imaging analysis are provided as supporting files.

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

## Acknowledgements

This manuscript is dedicated to Jürgen Langenbach, long-time science journalist at Die Presse, and to the memory of Prudence Dulormne. This research was funded by the Agence Nationale de la Recherche ANR GASTROMOVE - ANR-19-CE30-0016-01, by the Université de Paris IDEX Emergence en Recherche CHEVA19RDX-MEUP1, by the CNRS PEPS INSIS "COXHAM" grant, by the Labex "Who AM I?" ANR-11-LABX-0071, and by the Imaging platform BioEmergences-IBiSA, ANR-10-INBS-04 and ANR-11-EQPX-0029. We thank Sylvie Dufour for providing the Ht-PA::Cre mouse line, Ko Sugarawa for help with the Segment Anything Model under QuPath, Vincent Fleury, Michael Levin, Alexandre Ayed, Nathalie Rouach, Isabelle Arnoux, Olivier Romito and Master 2 students of the Université Paris Cité Biomedical Engineering 2024-2025 program for thoughtful discussions and/or performing experiments together.

## Author contributions

NRC led the project, obtained funding, performed experiments, analyzed data, synthesized data, wrote the draft and revised the paper; TS implemented new analysis methods and analyzed data; ZC performed experiments and analyzed data; NB, FG, MF, AEM, LC, MD, ILP, LZ performed experiments; LZ critically discussed the data; NB revised the draft.

## Competing interests

The authors declare no competing interests.
