## [Transparent Peer Review file · Nature Communications]

Endothelin-3 and T-type Ca²⁺ channels drive enteric neural crest cell calcium activity, contractility and migration

Corresponding Author: Dr Nicolas Chevalier

Version 0:

Reviewer comments:

Reviewer #1

(Remarks to the Author)

In this manuscript, Chevalier et al. analyzed in mouse embryos how calcium activity, regulated by EDN3/EDNRB signaling, influences the migration of enteric neural crest cells (ENCCs), and what is its role in the pathogenesis of Hirschsprung disease (HD)? The authors applied multiple organ cultures and migration assays combined with calcium activity analysis, calcium imaging for enteric neural crest cells, and immunocytochemistry to characterize the developing ENS. Greater understanding of the ENS and emphasizing the complex molecular and mechanobiological pathways during ENCC differentiation is a great value to both the congenital neurointestinal diseases and the enteric neural crest biology field. The presented findings provide mechanistic insight into how calcium waves are required for the ENCC migration and differentiation in the developing hindgut.

The manuscript's topic is interesting and important in the context of hindgut ENS development. However, there are several major concerns in the present manuscript.

Include anti-Anti-CaV3.2 (CACNA1H) staining or in situ hybridization to demonstrate the expression of Ca²⁺ channels on E11.5 wavefront ENCCs.

According to Hirst et al (2015) ENCCs do not require ion channel activity to migrate through the gut mesenchyme. Explain why your results is different.

If the 1mM ascorbic acid (specific CAV3.2 blocker) strongly reduced CA, how does the ascorbic acid affect ENCC migration? Why the effect of ascorbic acid on ENCC migration is missing from your results (see Fig. 3f).

Line 23: This statement is not supported by your results: "CaV3.2 channel blockade results in an ENCC migration defect". It is not proven that the HD-type aganglionosis was caused by the blockade of the CaV3.2 channel.

Did you see CA in transmesenteric ENCCs?

Line 58: typo "could at times"

Line 62: migration front versus more proximal regions terminology should be changed according to Stavely et al., 2023; Development and Zhou B et al., 2024 to wavefront (these are the leading front of the ENCCs) and trailing cells (these are the stationary ENCCs) see also Druckenbrod and Epstein, 2007.

Line 61-62: There is no apparent difference in the localization of CA and the immunoreactivity of Sox10 and Tuj1, likely due to technical difficulties with the Sox10 immunostaining. To better distinguish between Sox10 and Tuj1 expression, the authors should focus on the cecum, where the Sox10+/Tuj1- ENCC wavefront is located at this developmental stage.

Fig1: What is the red color on image "a"? Tuj1 or Sox10? Add to the legend.

Fig S3: add proper labeling (midgut, hindgut, cecum) to mark the orientation of the embryonic gut samples

Fig S3: Are the different E11.5 samples in the same orientation?

Line 82 and 85. Why was it important to change the concentration (10 fold) of EDN3 at stage E11.5 (1nM) to 10nM at E12.5?

Video 4: Can you mark the cells that retract their processes after BQ788 treatment?

Fig 2f: include high power images (inset) to show the morphology of the CTRL and BQ788-treated cells.

Line 128: add the effect of ascorbic acid to Fig 3f.

Video S9: The time-lapse movie is too short to see the backflow of mesenchyme.

Line 148: According to Barlow et al 2003 and Nagy and Goldstein 2006, EDN3 inhibits the migration of ENCCs in response to GDNF. Explain why your data using mouse ENCCs is different from the previously published results. What was the explant: midgut, ceca or hindgut?

Line 149 and Fig 3d. Include GDNF alone to prove the positive effect of 1nM EDN3+10ngGDNF on the ENCC migration distance.

Lines 169-170: The following sentence is not clear: "Several observations...."

Line 171: It is just a speculation that the effect of the CA blockers indicates a loss of ENCC-ECM traction.

Line 180-186: it is not related to the Results section.

Fig. 4. What are the green cells on a,b,c images? Are they isolated ENCCs or cells emigrated from the gut explants in response to GDNF?

Fig. 4: This figure is poorly presented in the Results section. a,b, c images are not introduced properly in the text.

line 201: "ENCCS"

To increase the quantity of their work, the authors should include a more sensitive ENCC culture assay (e.g., culture of isolated Wnt1-GCaMP 11.5 or 12.5 dpc mouse intestine, either with FACS sorting or induce to emigrate from the gut explant in response to GDNF) to study with high-power microscopy and prove that modulation of Ca²⁺ activity affects the ENCC migration and cell contractility. You may include phalloidin staining to mark the cell skeleton of the cytoplasmic processes.

Fig.5. it is not clear why the schematic figure is important for the Discussion section.

What is the relationship between β 1-integrin signalling pathway, Ca²⁺-dependent cell adhesion molecules and Endothelin 3 activation or inhibition during ENCC migration?

The aganglionic segment of EDN3 mutant human and mouse is restricted to the distal colorectum. Compare Ca²⁺ frequency of the trailing and wavefront ENCCs in control and ls/ls E12.5 (dpc) mouse hindgut to demonstrate that CA is critical for hindgut ENCC migration and distal colorectum ENS formation.

(Remarks on code availability)

-

Reviewer #2

(Remarks to the Author)

The manuscript by Chevalier et al is a well written and nicely illustrated description of a series of studies of murine embryonic gut explants which demonstrate a role for endothelin-3 (Edn3) / endothelin receptor B (Ednrb)-mediated calcium fluxes in the migratory activity of enteric neural crest cells (ENCCs). The study builds on previous data, appropriately cited in the manuscript, which indicate that calcium waves exist in ENCCs and that other members of the endothelin ligand / receptor family are known to influence cell contractility via calcium activation. The results provide novel insight into how Ednrb-signaling promotes colonization of the gut by ENCCs, discounts to some extent a prevailing counter hypothesis that the primary function of Endrb-signaling is to suppress differentiation and promote proliferation, and may lead to new approaches to Hirschsprung disease and other disorders of neural crest migration (e.g., piebaldism, melanoma, neuroblastoma).

The following concerns should be considered:

1. Abstract, line 18 should read "... glia-derived neurotrophic factor (GDNF) / Rearranged in Transfection receptor tyrosine kinase (RET) and the ..."
2. Abstract, 22: "triphosphate-sensitive" (insert hyphen)
3. Abstract, line 23: "EDNRB, Cl- channel, or CaV3.2-channel" (need to insert "channel" because EDNRB is not a channel)
4. Main Text, line 39: "Vagal-derived" (insert hyphen)
5. Main Text, line 42: "1:5000 births" is missing the "s". It is probably not correct to claim that HD is the most frequent

neurocristopathy. Neurofibromatosis is 1:3000. Velocardiofacial syndrome is 1:3000. Consider stating “one of the most frequent neurocristopathies”.

6. The Abstract and line 55 of the Main Text should indicate that this study was based on mice. For line 55, it would suffice to write “... ex-vivo embryonic mouse guts ...”

7. Main Text, line 60: Should read “ $7.5 \times 10^{-4} \pm 2 \times 10^{-4}$ ” (introduce missing X's)

8. Main Text, line 93: Technically, the *Is* allele is not a “knock-out” per se. The missense mutation predicts a protein that cannot be cleaved into mature Edn3. Functionally this is equivalent to complete loss of mature Edn3, but not abrogation of the unprocessed precursor protein which conceivably might have other functions.

9. Figure 1B and related data showing differences in Ca^{++} activity between E11.5 and E12.5 are fascinating and may warrant some contextual discussion in the manuscript. Specifically, the published work of Myun Shin and colleagues (Nature 1999;402:496; Nature Genet 2004;36:735), Coventry et al (Lab Invest 1994;71:82), and Drukenbrod Dev Biol 2005;287:125) collectively suggest that ENCC colonization of the cecum from the distal ileum may operate differently and possibly present a “developmental hurdle” which endothelin signaling helps negotiate. Whether endothelin signaling is operative after this hurdle is traversed is an open question, but the reduced Ca^{++} activity demonstrated between E11.5 and E12.5 lends some support to a model in which endothelin signals are most (solely?) important up until the ileocecum is traversed. Similarly, the seemingly dramatic change in response to exogenous Edn3 from “variable effects” at E11.5 to “an immediate increase of intracellular Ca^{++} across all ENCC's” at E12.5 seems to correlate with some stage-specific change in the responsiveness of these cells, which correlates temporally with their crossing the cecum.

10. Main Text, line 117 and Figure S9 and Figure S9 legend): It would be preferable to use the term “administration” or “introduction” in place of “injection”. The latter suggests something introduced into the cells or tissue directly.

11. Main Text, line 118: “... in the presence of EDTA (Fig.3e, ...” (EDTA is not shown in Figure 3e).

12. Discussion, line 193: It seems inconsistent to regard *Is/Is* mice as *edn3* “knock-outs” (see prior comment) and then state that they were first developed in the 1990s. As the authors recognize, the natural mutants of *edn3* and *ednrb* were “developed” by mouse fanciers many decades earlier.

13. Discussion, line 271: “Melanomas are” or “Melanoma is”.

14. Discussion, last two sentences: The potential relevance of the current study to neural crest-related cancer is important. In addition, or perhaps in place of, melanoma, the authors might consider mentioning neuroblastoma, a pediatric malignancy arguably closer in cellular phenotype to ENCCs and with evidence that calcium signaling correlates with prognosis (see Lange I, Koster J, Kooma DT. Calcium signaling regulates fundamental processes involved in Neuroblastoma progression. Cell Calcium. 2019 Sep;82:102052. doi: 10.1016/j.ceca.2019.06.006. Epub 2019 Jun 17. PMID: 31306997.)

15. Figures S1 and S2: It would be helpful to modify the legend of Figure S1 to include the same terminology and abbreviations as used throughout the other figures. If I understand correctly, the minimum peak prominence was threshold for intensity and minimum peak-to-peak distance of 3 sec is for frequency threshold. Also, the last line of the legend should read “automated”, not “automatized.” IR is not defined in the legend for Figure S2.

16. Figure S4 legend: Move space to read “Figure S4”. In the last line write “indicated (p values and red stars), unless ...”

(Remarks on code availability)

Code was not part of this study.

Reviewer #3

(Remarks to the Author)

The authors present a technically rigorous study demonstrating that calcium activity in the embryonic mouse gut is partly driven by EDNRB signaling, with further mechanistic dissection implicating chloride channels and CaV3.2 (T-type) calcium channels. While the experimental detail and imaging approaches are impressive, the overall motivation of the study is insufficiently articulated.

If the primary goal is to understand how calcium signaling is altered in the context of Hirschsprung disease (HSCR), additional data are needed from genetic models with known HSCR-associated mutations. The brief analysis in *Edn3^{Is/Is}* mice is a good start, but all subsequent experiments are performed in wild-type mice using pharmacological blockers. This limits the study's ability to mechanistically connect altered calcium dynamics to disease-relevant phenotypes.

More broadly, while the EDN3/EDNRB pathway's role in ENS development is well-established, and the identification of CaV3.2's involvement in ENCC bioelectric activity is novel, the functional link between CaV3.2 activity and HSCR pathogenesis remains speculative. The reliance on pharmacological inhibition without complementary genetic models (e.g., *Cacna1h* mutants, conditional knockouts) weakens causal interpretation. Likewise, the reported association of CACNA1H variants in HSCR is not supported by functional data or penetrance analyses.

To strengthen the manuscript and better position it within a disease framework, the authors should consider the following:

1. Poorly Articulated Link to Hirschsprung Disease

- The motivation for the study in the context of HSCR is weak and underdeveloped. While the authors state that EDN3/EDNRB and CACNA1H are involved in HSCR, they fail to:
 - o Provide a clear hypothesis about how Ca^{2+} signaling defects cause aganglionosis.
 - o Demonstrate in vivo relevance using proper genetic models with defined HSCR penetrance.

2. Inadequate Validation of the CACNA1H-HSCR Link

- The claim that CACNA1H SNPs contribute to HSCR is made based on one prior EWAS reference (Tang et al. 2017) without showing:

- o Whether this variant is functional (e.g., reduces CaV3.2 current or expression).
 - o Whether CaV3.2 deficiency alone recapitulates aganglionosis (and they acknowledge the knockout mouse is viable).
 - o If CACNA1H loss leads to reduced ENCC migration in vivo, not just in culture.
3. Heavy Reliance on Pharmacological Inhibition Without Adequate Genetic Validation
- Most of the mechanistic claims (e.g., CaV3.2 dependence) are based on pharmacology (Z944, ascorbic acid), which:
 - o Are not entirely specific (Z944 also blocks CaV3.1).
 - o May affect cell health (shown for some drugs).
 - No CaV3.2-specific knockdown/knockout experiments are provided to confirm specificity.
 - No rescue experiments (e.g., with CaV3.2 re-expression or EDNRB overexpression) are included to test causality.
4. Migration Assays Lack Rigorous Quantification
- The migration distances are modest and not accompanied by:
 - o ENCC cell number quantification—is reduced migration due to fewer cells?
 - o Proliferation or apoptosis assays to rule out cytotoxic effects of the drugs.
 - o In vivo rescue or dose-response studies to support pharmacological conclusions.

5. the rationale for doses of EDN3 added is not well explained- eg it says "1 nM EDN3 at E11.5 At stage E12.5, application of 10 nM EDN3" How were the doses calculated?

6. some of the statements are incorrect- 'EDN3 gene knockout model that develops Hirschsprung disease, the ls/ls mouse". ls/ls is a Ednrb deficient mice not End3 knockout

(Remarks on code availability)

There is no code involved

Version 1:

Reviewer comments:

Reviewer #1

(Remarks to the Author)

Thank you for clearly addressing each of the comments. I have no further questions or remarks.

(Remarks on code availability)

Reviewer #2

(Remarks to the Author)

The revised version of this manuscript adequately addresses all of the concerns raised in my original review. I remain impressed with the findings which collectively support the hypothesis that asynchronous Ca⁺-flux mediated contractions of enteric neural crest cells are influenced by EDN3/EDNRB signaling and mediated by CaV3 and Cl⁻ channels. I believe this is significant because, as the authors point out, it unifies the functions of EDNRB signaling in ENCCs with a more established role in mature smooth muscle contractile regulation. I have no strong reservations but offer the following comments:

1. The immunohistochemistry results shown in Figure 3 seem to show similar diffuse labeling of all the cells. While this may reflect specific labeling with each of the antibodies used, it would be helpful to show that some cell type known not to express the CaV protein of interest is immunonegative.

2. I have trouble following the logic of this sentence (line 308): "We note that the CaV2^{-/-} mouse is viable, although the mutation induces prenatal lethality." Is the intended meaning fetal versus post-natal viability. Please clarify.

Minor typo: line 72: ENCC-specific (add hyphen)

(Remarks on code availability)

Reviewer #3

(Remarks to the Author)

All my concerns have been addressed. I don't have any additional comments

(Remarks on code availability)

There is no code associated with the manuscript

We thank the referees for their constructive criticism and appreciation of our work. We have now backed up our investigation with further experimental data and clarified points of concern, you will find point-by-point answers in green, with modification to the manuscript in italic.

Reviewer #1 (Remarks to the Author):

In this manuscript, Chevalier et al. analyzed in mouse embryos how calcium activity, regulated by EDN3/EDNRB signaling, influences the migration of enteric neural crest cells (ENCCs), and what is its role in the pathogenesis of Hirschsprung disease (HD)? The authors applied multiple organ cultures and migration assays combined with calcium activity analysis, calcium imaging for enteric neural crest cells, and immunocytochemistry to characterize the developing ENS. Greater understanding of the ENS and emphasizing the complex molecular and mechanobiological pathways during ENCC differentiation is a great value to both the congenital neurointestinal diseases and the enteric neural crest biology field. The presented findings provide mechanistic insight into how calcium waves are required for the ENCC migration and differentiation in the developing hindgut.

The manuscript's topic is interesting and important in the context of hindgut ENS development. However, there are several major concerns in the present manuscript.

Include anti-Anti-CaV3.2 (CACNA1H) staining or in situ hybridization to demonstrate the expression of Ca²⁺ channels on E11.5 wavefront ENCCs.

We have performed additional experiments and now include IHCs for CaV3.1, CaV3.2 and CaV3.3 (new Fig.3). To our surprise ENCCs were positive for all three channel types, not just CaV3.2, as we had initially assumed based on the PCR data of Hirst et al. We therefore further tested whether SAK3, the only commercially available CaV3 agonist (it amplifies CaV3.1 and CaV3.3 currents), could enhance Ca²⁺ activity. We found that it does and added the results in Fig.3b, Video S6 and Fig.S10e. We further found (new Fig.4) that SAK3 could potentiate the effect of GDNF in the collagen gel migration assay, much like EDN3 – a noteworthy result. We finally investigated whether SAK3 could “rescue” CA inhibition by BQ788: although it did increase CA, it remained at very low levels (~0.1 events/min/100 μm²) compared to native E11.5 guts (Fig.S14a,b). It also did not improve ENCC migration down the colon when applied together with BQ (new Fig.5).

We note that the CaV3 antibodies (Alomone Labs) worked well on ENCCs that had migrated from explants, i.e., without embedding, freezing or cutting of the samples. On the contrary, we could not detect specific signal from any of the 3 antibodies when the same IHC protocol was applied to frozen slices of native E11.5 gut. We describe this in the Materials & Methods.

We have summarized these findings in the revised version, indicating all changes to the initial version of the manuscript in green. Thank you for suggesting this experiment, which rectifies our conclusions and puts us on the right track for future research.

According to Hirst et al (2015) ENCCs do not require ion channel activity to migrate through the gut mesenchyme. Explain why your results is different.

We had addressed these differences previously but have now grouped them in a single paragraph of the discussion section to clarify:

“T-type receptor blockade by mibefradil 1 μ M was previously reported not to affect ENCC migration²⁸. It is probable that this concentration was insufficient to cause aganglionosis, because the IC50 of mibefradil on isolated cells is 3 μ M⁵⁵; in our experience, effective, sustained blocking of channel activity in whole-gut cultures requires concentrations far above drug IC50 to yield a migration phenotype. Hirst et al.²⁸ also blocked Cf channels with NPPB 100 μ M from an only 100-fold stock solution in a non-specified solvent, and found no effect on migration or on ENCC survival. In our experiments, NPPB 100 μ M disrupted CA and migration by systematically inducing ENCC death (n=11/11, Fig.S15). The choice of NPPB solvent or inconsistencies in the actual concentrations applied by Hirst et al. may explain the discrepancy. More generally, we were guided in our choices of pharmacological compounds and concentrations by the Ca²⁺ response, whereas Hirst et al. applied these compounds “blindly” to wildtype guts.”

If the 1mM ascorbic acid (specific CAV3.2 blocker) strongly reduced CA, how does the ascorbic acid affect ENCC migration? Why the effect of ascorbic acid on ENCC migration is missing from your results (see Fig. 3f).

We had just considered the immediate effect on Ca²⁺ activity of ascorbic acid. We have now added n=7 samples with tracking of short- and long-term effects on Ca²⁺ activity (Fig.3b, Fig.S10d, Fig.S12) and its effect on migration (Fig.5): we find that it significantly decreases the migration distance covered by the ENCCs at J+1 compared to controls, although to a lesser degree than Z944, NFA or BQ.

Line 23: This statement is not supported by your results: “CaV3.2 channel blockade results in an ENCC migration defect”. It is not proven that the HD-type aganglionosis was caused by the blockade of the CaV3.2 channel.

Indeed, we have changed the statement to “T-type channel blockade ...” and wherever relevant in the manuscript.

Did you see CA in transmesenteric ENCCs?

Yes we observe CA in transmesenteric ENCCs as well, we now point them out in Fig.S3 and in the results section I.71 “CA was also present in transmesenteric ENCCs (Fig.S3).”

Line 58: typo “could at times”

We modified : “Ca²⁺ transients could sometimes propagate ...”

Line 62: migration front versus more proximal regions terminology should be changed according to Stavely et al., 2023; Development and Zhou B et al., 2024 to wavefront (these are the leading front of the ENCDCs) and trailing cells (these are the stationary ENCDCs) see also Druckenbrod and Epstein, 2007.

We modified this l.70 “it was significantly higher at the ENCC wavefront located at the ileo-cecal junction (ilcc) than in trailing ENCCs (Fig.1c), ...”

Line 61-62: There is no apparent difference in the localization of CA and the immunoreactivity of Sox10 and Tuj1, likely due to technical difficulties with the Sox10 immunostaining. To better distinguish between Sox10 and Tuj1 expression, the authors should focus on the cecum, where the Sox10+/Tuj1- ENCDC wavefront is located at this developmental stage.

We had an issue with washing of the antibodies leading to these red aggregates. We repeated this experiment, the new Video S2 shows clearly that CA correlates with Sox10, not with Tuj1.

Fig1: What is the red color on image “a”? Tuj1 or Sox10? Add to the legend.

We had specified in the legend : “Red: CA frequency heatmap shows concentration of activity at the cecum and the ileum anti-mesenteric border (see also Fig.S3).” We used a finer spatial mesh (compared to the 250 μm² mesh used in the remainder of the report, see for example Fig.S1, S3) specifically for this image, for presentation purposes.

Fig S3: add proper labeling (midgut, hindgut, cecum) to mark the orientation of the embryonic gut samples

Fig S3: Are the different E11.5 samples in the same orientation?

No were are not all in the same orientation ; we corrected this for clarity and added labels

Line 82 and 85. Why was it important to change the concentration (10 fold) of EDN3 at stage E11.5 (1nM) to 10nM at E12.5?

We tried both 1 and 10 nM at E12.5, but mentioned only the effect at 10 nM because it differed qualitatively from that observed at E11.5. We now include the data on 1 nM at E12.5 as well in Fig.S5c and commented:

“In stage E12.5 ileum, CA also increased threefold at 1 nM EDN3 (Fig.S5), and the application of 10 nM EDN3 induced an immediate increase of intracellular Ca²⁺ across all ENCCs (Fig.2b, Video S3, Fig.S5).”

Video 4: Can you mark the cells that retract their processes after BQ788 treatment?

Fig 2f: include high power images (inset) to show the morphology of the CTRL and BQ788-treated cells.

We added arrows to Video 4 showing the retracting cells, and zoomed-in images of cell morphologies in Fig.2f.

Line 128: add the effect of ascorbic acid to Fig 3f.

We added this as stated previously.

Video S9: The time-lapse movie is too short to see the backflow of mesenchyme.

The video needs to be loaded in ImageJ and the time-cursor tracked fast-forward& backward repeatedly between frames 70 and 89, focusing on the areas indicated by the arrow; we clarified this in the Supplementary Material description. You will see that the mesenchyme moves in the opposite direction to the ENCCs.

Line 148: According to Barlow et al 2003 and Nagy and Goldstein 2006, EDN3 inhibits the migration of ENCCs in response to GDNF. Explain why your data using mouse ENCCs is different from the previously published results. What was the explant: midgut, ceca or hindgut?

We have summarized our thoughts on this in this new paragraph of the discussion section:

“We found that GDNF and EDN3 (or SAK3) have a synergistic action on ENCC migration from explants (Fig.3). Our results are in agreement with the findings of Bergeron et al.⁶⁰. We did not observe that EDN3 and GDNF were antagonistic, as has been reported in other studies with mouse²⁵, rat⁶¹ and chicken embryos⁴⁶. These studies were performed respectively at stages E10.5-E11, E13 and E8, at EDN3 concentrations of 100, 20 and 100 nM and after 16h, 2-3 days and 3 days of migration. It is difficult to fathom the reasons of this discrepancy but we stress that: 1°) migration after 16h²⁵ is too scarce to be precisely quantified, 2°) the EDN3 concentrations applied by other investigators are 1-2 orders of magnitude higher than the maximum pro-CA and pro-migratory effect we report at 1 nM. 1 nM likely reflects the concentration encountered physiologically by ENCCs as they migrate down the gut mesenchyme. We found that CA was not significantly increased at 10 nM EDN3 compared to 1 nM (Fig.2a, Fig.S5c), indicating a saturation effect. Administration of EDN3 10 nM at E12.5 gave rise to a single pan-ENCC transient, which, although impressive, most likely never occurs physiologically.”

Line 149 and Fig 3d. Include GDNF alone to prove the positive effect of 1nM EDN3+10ngGDNF on the ENCC migration distance.

You surely meant edn3 alone (GDNF alone was already presented). We have performed new experiments and added the result to the new Fig.4. Edn3 alone leads to a small, but significant increase in migration, although not as marked as GDNF alone. This is consistent with the findings of Bergeron et al.

Lines 169-170: The following sentence is not clear: “Several observations....”

We changed the punctuation and numbered the 3 observations to make it clearer.

“Several observations indicated that Ca²⁺ oscillations could promote migration by enhancing cell contractility, as in smooth muscle: 1°) ...; 2°) ...; 3°) ... “

Line 171: It is just a speculation that the effect of the CA blockers indicates a loss of ENCC-ECM traction.

We now include new traction force data (new Fig.6) showing that the T-type Ca²⁺ channel antagonist Z944 decreases the ENCC-ECM traction force similarly to BQ788.

Line 180-186: it is not related to the Results section.

We have moved this to the Discussion section

Fig. 4. What are the green cells on a,b,c images? Are they isolated ENCCs or cells emigrated from the gut explants in response to GDNF?

Fig. 4: This figure is poorly presented in the Results section. a,b, c images are not introduced properly in the text.

The green cells (now in Fig.6a) are ENCCs emigrating from the explant (to the left, not seen in the field-of-view). We have clarified this in the text

l.220 “To quantify the force exerted by ENCCs on the ECM, we let ENCCs migrate from E11.5 midgut explants in a collagen gel seeded with fluorescent beads that served as fiducial markers of gel deformation (Fig.6a inset).”

and in the legend of Fig.6:

“ENCC 3D migration from a E11.5 midgut explant (to the left, not in the field of view)”

and have added an inset scheme of the experiment in Fig.6a.

line 201: “ENCCS”

We corrected this

To increase the quantity of their work, the authors should include a more sensitive ENCC culture assay (e.g., culture of isolated Wnt1-GCaMP 11.5 or 12.5 dpc mouse intestine, either with FACS sorting or induce to emigrate from the gut explant in response to GDNF) to study with high-power microscopy and prove that modulation of Ca²⁺ activity affects the ENCC

migration and cell contractility. You may include phalloidin staining to mark the cell skeleton of the cytoplasmic processes.

Perhaps it was not clear from the previous description of Fig.6, but that is precisely what we did (culture of isolated Wnt1-GCaMP 11.5 dpc mouse intestine induced to emigrate from the gut explant in response to GDNF). The inclusion of beads in the gel and monitoring of their displacements to assess traction forces is novel in the enteric neural crest field.

Fig.5. it is not clear why the schematic figure is important for the Discussion section.

This figure (now Fig.7) is important because it summarizes the molecular route we have uncovered going from edn3 docking to EDNRB, to Ca²⁺ release and traction force generation necessary for migration.

What is the relationship between β 1-integrin signalling pathway, Ca²⁺-dependent cell adhesion molecules and Endothelin 3 activation or inhibition during ENCC migration?

β 1-integrin link the cell cytoskeleton to the ECM, and thus allow the transmission of contractile forces generated by the ENCCs to the ECM. ENCC specific knock-out of β 1-integrin induces a Hirschsprung phenotype (Breau et al. 2006, Gazquez et al. 2016, Chevalier et al. 2021). Edn3 promotes Ca²⁺ activity, which generates ENCC contractility (Fig.6). We do not discuss Ca²⁺-dependent cell adhesion molecules (cadherins) in the manuscript, beyond the observation that EDTA obviously perturbs them (Video S5).

I.264: "ENCC contraction leads to an increased traction force to the extracellular-matrix, that is necessary for their migration inside the gut mesenchyme. This force is the reason why investigators have found it indispensable to pin⁴⁵ or embed the gut tract during ex-vivo migration assays⁴⁶, as it otherwise leads to tissue shrinkage and improper migration. This force is transmitted via β 1-integrins, and ENCC-specific β 1-integrin mutants have been shown to present with a HD phenotype⁴⁷⁻⁴⁹."

The aganglionic segment of EDN3 mutant human and mouse is restricted to the distal colorectum. Compare Ca²⁺ frequency of the trailing and wavefront ENCCs in control and ls/ls E12.5 (dpc) mouse hindgut to demonstrate that CA is critical for hindgut ENCC migration and distal colorectum ENS formation.

We compared the frequency between trailing and wavefront ENCCs in control samples (Fig.1c). We cannot do the same at present for ls/ls, as we do not have a mouse line that presents both GCaMP and the ls/ls mutation.

Reviewer #1 (Remarks on code availability):

Reviewer #2 (Remarks to the Author):

The manuscript by Chevalier et al is a well written and nicely illustrated description of a series of studies of murine embryonic gut explants which demonstrate a role for endothelin-3 (Edn3) / endothelin receptor B (Ednrb)-mediated calcium fluxes in the migratory activity of enteric neural crest cells (ENCCs). The study builds on previous data, appropriately cited in the manuscript, which indicate that calcium waves exist in ENCCs and that other members of the endothelin ligand / receptor family are known to influence cell contractility via calcium activation. The results provide novel insight into how Ednrb-signaling promotes colonization of the gut by ENCCs, discounts to some extent a prevailing counter hypothesis that the primary function of Ednrb-signaling is to suppress differentiation and promote proliferation, and may lead to new approaches to Hirschsprung disease and other disorders of neural crest migration (e.g., piebaldism, melanoma, neuroblastoma).

The following concerns should be considered:

1. Abstract, line 18 should read "... glia-derived neurotrophic factor (GDNF) / Rearranged in Transfection receptor tyrosine kinase (RET) and the ..."
2. Abstract, 22: "trisphosphate-sensitive" (insert hyphen)
3. Abstract, line 23: "EDNRB, Cl⁻ channel, or CaV3.2-channel" (need to insert "channel" because EDNRB is not a channel)
4. Main Text, line 39: "Vagal-derived" (insert hyphen)
5. Main Text, line 42: "1:5000 births" is missing the "s". It is probably not correct to claim that HD is the most frequent neurocristopathy. Neurofibromatosis is 1:3000. Velocardiofacial syndrome is 1:3000. Consider stating "one of the most frequent neurocristopathies".
6. The Abstract and line 55 of the Main Text should indicate that this study was based on mice. For line 55, it would suffice to write "... ex-vivo embryonic mouse guts ..."

Thank you, we have corrected all these points, and also had to shorten the abstract to <150 words.

7. Main Text, line 60: Should read " $7.5 \times 10^{-4} \pm 2 \times 10^{-4}$ " (introduce missing X's)
8. Main Text, line 93: Technically, the Is allele is not a "knock-out" per se. The missense mutation predicts a protein that cannot be cleaved into mature Edn3. Functionally this is equivalent to complete loss of mature Edn3, but not abrogation of the unprocessed precursor protein which conceivably might have other functions.

We corrected this:

"We finally tested whether CA differed in an EDN3 missense mutation model that develops Hirschsprung disease ..."

9. Figure 1B and related data showing differences in Ca⁺⁺ activity between E11.5 and E12.5 are fascinating and may warrant some contextual discussion in the manuscript. Specifically, the

published work of Myun Shin and colleagues (Nature 1999;402:496; Nature Genet 2004;36:735), Coventry et al (Lab Invest 1994;71:82), and Drukenbrod Dev Biol 2005;287:125) collectively suggest that ENCC colonization of the cecum from the distal ileum may operate differently and possibly present a “developmental hurdle” which endothelin signaling helps negotiate. Whether endothelin signaling is operative after this hurdle is traversed is an open question, but the reduced Ca⁺⁺ activity demonstrated between E11.5 and E12.5 lends some support to a model in which endothelin signals are most (solely?) important up until the ileocecum is traversed. Similarly, the seemingly dramatic change in response to exogenous Edn3 from “variable effects” at E11.5 to “an immediate increase of intracellular Ca⁺⁺ across all ENCC’s” at E12.5 seems to correlate with some stage-specific change in the responsiveness of these cells, which correlates temporally with their crossing the cecum.

We understand your thoughts, and point out the following:

- We find that CA inhibitors slow down migration between E11.5 and E11.5+1 (Fig.5). ENCCs at E11.5 are at the ileo-cecal junction and migrate through the cecum and then the colon. It is possible that most of the slowing down linked to CA occurs in the cecum, and that for this reason the ENCC only barely reach the colon; is it equally possible that slowing down occurs both in the cecum and colon, we cannot tell from our data.
- Edn3 expression is highest in the cecum, and there is little expression in the hindgut at E12 (in-situ of Leibl et al., 1999), which indeed correlates with the lower CA we measure at the wavefront at E12.5 (compared to E11.5); this indeed supports a model where edn3 is most necessary up until the cecum is traversed. Aganglionosis of the colon can result from slowing down of the migration front at any space or time-point, not necessarily from slower migration of the ENCCs specifically in the colon.
- Concerning the change “from “variable effects” at E11.5 to “an immediate increase of intracellular Ca⁺⁺ across all ENCC’s” at E12.5”, we stress that the effect of edn3 1nM at E11.5 is almost systematic (10/12 samples) and quite dramatic (3-fold CA increase on average). We do not know why edn3 10 nM induces a pan-ENCC Ca²⁺ wave at E12.5, but not at E11.5. We doubt that it is related to ENCCs crossing the cecum, because these pan-ENCC waves were found in regions proximal to the cecum (ileum but also jejunum). It doesn’t seem to be related to a change in the electric (gap-junctional) connectivity of the ENCC network because pan-ENCC Ca²⁺ waves could be induced by ATPe at E11.5 (Video S8). It doesn’t seem to be related to increased ion channel expression because spontaneous CA decreased with increasing age. EDNRB receptors at E12.5 may be less saturated with endogenous edn3, and therefore react more sharply in response to exogenous edn3.

We have summarized some of these thoughts in the discussion:

“2°) the EDN3 concentrations applied by other investigators are 1-2 orders of magnitude higher than the maximum pro-CA and pro-migratory effect we report at 1 nM. 1 nM likely reflects the concentration encountered physiologically by ENCCs as they migrate down the gut mesenchyme. We found that CA was

not significantly increased at 10 nM EDN3 compared to 1 nM (Fig.2a, Fig.S5c), indicating a saturation effect. Administration of EDN3 10 nM at E12.5 gave rise to a single pan-ENCC transient, which, although impressive, most likely never occurs physiologically. The pan-ENCC Ca²⁺ surge induced by exogenous EDN3 may occur only at E12.5 (and not at E11.5) because of a reduced saturation of EDNRB receptors by endogenous EDN3 at this stage, making it more sensitive to exogenous application.

Edn3 expression is highest in the cecum, with little expression in the hindgut at E12²⁶, which correlates with reduced CA at the ENCC wavefront at E12.5 (Fig.1). These observations suggest that EDN3 may be most necessary up until the cecum is traversed; slowing down of ENCCs by CA activity inhibitors in our ex-vivo assay (Fig.5) may have primarily affected migration through the cecum, resulting in a paucity of ENCCs in the colon. Colonic aganglionosis can result from slowed-down migration at any point of their journey down the gut, not necessarily from slower migration in the colon. "

10. Main Text, line 117 and Figure S9 and Figure S9 legend): It would be preferable to use the term "administration" or "introduction" in place of "injection". The latter suggests something introduced into the cells or tissue directly.

We replaced the word "injection" by "administration" or "introduction" where relevant.

11. Main Text, line 118: "... in the presence of EDTA (Fig.3e, ..." (EDTA is not shown in Figure 3e).

We removed the reference to Fig.3e.

12. Discussion, line 193: It seems inconsistent to regard ls/ls mice as edn3 "knock-outs" (see prior comment) and then state that they were first developed in the 1990s. As the authors recognize, the natural mutants of edn3 and ednrb were "developed" by mouse fanciers many decades earlier.

We agree, we rephrased this as such:

"As the roles of endothelins in blood pressure regulation became more prominent, mutant mice for EDN3^{41,42} and EDNRB⁴³ mouse were re-examined"

13. Discussion, line 271: "Melanomas are" or "Melanoma is".

We corrected this

14. Discussion, last two sentences: The potential relevance of the current study to neural crest-related cancer is important. In addition, or perhaps in place of, melanoma, the authors might consider mentioning neuroblastoma, a pediatric malignancy arguably closer in cellular

phenotype to ENCCs and with evidence that calcium signaling correlates with prognosis (see Lange I, Koster J, Koomoa DT. Calcium signaling regulates fundamental processes involved in Neuroblastoma progression. Cell Calcium. 2019 Sep;82:102052. doi: 10.1016/j.ceca.2019.06.006. Epub 2019 Jun 17. PMID: 31306997.)

Thank you, we did not know about this literature, it is obviously relevant, we have added the following references:

“Melanoma⁷ and neuroblastoma⁵⁹ are known to recapitulate many NCC traits. T-type Ca²⁺ channel upregulation is associated with melanoma aggressiveness⁶⁰ while Ca²⁺ signaling is altered in neuroblastoma⁶¹”

15. Figures S1 and S2: It would be helpful to modify the legend of Figure S1 to include the same terminology and abbreviations as used throughout the other figures. If I understand correctly, the minimum peak prominence was threshold for intensity and minimum peak-to-peak distance of 3 sec is for frequency threshold. Also, the last line of the legend should read “automated”, not “automatized.” IR is not defined in the legend for Figure S2.

We have added the missing abbreviations in the legend of Fig.S1 and S2. Yes your understanding of the thresholds is correct.

16. Figure S4 legend: Move space to read “Figure S4”. In the last line write “indicated (p values and red stars), unless ...”

We have corrected this

Reviewer #2 (Remarks on code availability):

Code was not part of this study.

Reviewer #3 (Remarks to the Author):

The authors present a technically rigorous study demonstrating that calcium activity in the embryonic mouse gut is partly driven by EDNRB signaling, with further mechanistic dissection implicating chloride channels and CaV3.2 (T-type) calcium channels. While the experimental detail and imaging approaches are impressive, the overall motivation of the study is insufficiently articulated.

If the primary goal is to understand how calcium signaling is altered in the context of Hirschsprung disease (HSCR), additional data are needed from genetic models with known HSCR-associated mutations. The brief analysis in *Edn3^{ls/ls}* mice is a good start, but all subsequent experiments are performed in wild-type mice using pharmacological blockers. This limits the study's ability to mechanistically connect altered calcium dynamics to disease-relevant phenotypes.

More broadly, while the EDN3/EDNRB pathway's role in ENS development is well-established, and the identification of CaV3.2's involvement in ENCC bioelectric activity is novel, the functional link between CaV3.2 activity and HSCR pathogenesis remains speculative. The reliance on pharmacological inhibition without complementary genetic models (e.g., *Cacna1h* mutants, conditional knockouts) weakens causal interpretation. Likewise, the reported association of CACNA1H variants in HSCR is not supported by functional data or penetrance analyses.

We thank the reviewer for the overall impression and comments. We have reformulated the manuscript to make our motivation clearer: our primary aim was not to understand how calcium signaling is altered in Hirschsprung disease, but to understand the role and molecular mechanisms of calcium signaling during physiological migration. This required the establishment of new methodological tools (Ca²⁺ imaging and analysis protocol, 3D traction force assay). Our conclusions pertaining to Hirschsprung disease are perspectives resulting from this work, which motivate further investigations with dedicated mouse models as you point out, for future research projects. The mostly-pharmacological investigation we present is a necessary first step before moving on to the more financially and time-demanding developments of conditional knockouts.

This is only the second study to address Ca²⁺ signaling in enteric neural crest cells, after the seminal first observation of these transients by Hao et al. in 2017. We have added a wealth of information:

- Most of this activity occurs as single-cell events, and not as multicellular waves. The focus on multicellular waves in Hao et al.'s work did not allow them to grasp the fundamental mechanisms of ENCC electrical activity, nor its precise spatial and temporal dependence (Fig.1).
- We revealed that *edn3*/EDNRB (Fig.2), T-type Ca²⁺ channels and Cl⁻ channels (Fig.3) are the main drivers of Ca²⁺ activity; that purinergic signaling, which was presented as the main source of Ca²⁺ waves in Hao et al.'s work, is merely a modulator. We also now point

out more clearly in the discussion why the pharmacological investigation of Hirst et al. (2015) on the role of ion channels in ENCC migration overlooked the major role played by T-type Ca^{2+} channels and Cl^- channels.

- We now understand that this Ca^{2+} activity translates to increased cell contractility (Fig.6), which is necessary for their migration down the gut mesenchyme (Fig.4&5). Previous investigations had failed to establish a clear link between Ca^{2+} activity and migration. The 3D traction force assay we present in Fig.6 is the first to be applied in the enteric neural crest field and the results are, in our opinion, fascinating, because they explain the fundamental biophysical mechanism by which *edn3/EDNRB* enhances ENCC migration.

We reflected these considerations in the manuscript in the abstract (mildering our claim regarding HD and *CACNA1H*, and also shortening it to get below the 150 word limit), introduction (l.52-60), discussion (l.291-306), outlining all changes in green.

To strengthen the manuscript and better position it within a disease framework, the authors should consider the following:

1. Poorly Articulated Link to Hirschsprung Disease

- The motivation for the study in the context of HSCR is weak and underdeveloped. While the authors state that *EDN3/EDNRB* and *CACNA1H* are involved in HSCR, they fail to:

- o Provide a clear hypothesis about how Ca^{2+} signaling defects cause aganglionosis.

We clarified this in the discussion section, and our argument is further bolstered by new measurements showing that Z944 decreases the traction force (new Fig.6c,e):

*l.260: "We reveal here that the immediate action of EDN3 on ENCCs is in fact nearly identical to its effect on smooth muscle tone: it triggers Ca^{2+} activity (Fig.1,2), via a very similar molecular route (Fig.3,7) to vSMC, and the increased cytosolic Ca^{2+} oscillations enhance migration (Fig.4,5) by inducing cell contractility (Fig.6). vSMC contraction leads to vessel constriction; ENCC contraction leads to an increased traction force to the extracellular-matrix, that is necessary for their migration inside the gut mesenchyme. This force is the reason why investigators have found it indispensable to *pin*⁴⁵ or embed the gut tract during ex-vivo migration assays⁴⁶, as it otherwise leads to tissue shrinkage and improper migration. The traction force is transmitted via $\beta 1$ -integrins, and ENCC-specific $\beta 1$ -integrin mutants have been shown to present with a HD phenotype⁴⁷⁻⁴⁹. Ca^{2+} signaling defects reduce the traction force of the ENCCs to the extracellular matrix, lowering their migration speed, resulting in colonic aganglionosis. Our investigation shows that ENCCs are akin to miniature muscles that contract and crawl in response to a constrictor peptide, endothelin 3."*

- o Demonstrate in vivo relevance using proper genetic models with defined HSCR penetrance.

This will be investigated in a future project.

2. Inadequate Validation of the CACNA1H-HSCR Link

- The claim that CACNA1H SNPs contribute to HSCR is made based on one prior EWAS reference (Tang et al. 2017) without showing:
 - o Whether this variant is functional (e.g., reduces CaV3.2 current or expression).
 - o Whether CaV3.2 deficiency alone recapitulates aganglionosis (and they acknowledge the knockout mouse is viable).

We have now added n=7 samples cultured with the specific CaV3.2 blocker ascorbic acid, with tracking of short- and long-term effects on Ca²⁺ activity (Fig.3b, Fig.S10d, Fig.S12) and migration (Fig.5): we found that ascorbic acid significantly decreases the migration distance covered by ENCCs at J+1 compared to controls, although to a lesser degree than Z944, NFA or BQ.

Following comments by another reviewer, we have added IHCs for CaV3.1, Cav3.2 and CaV3.3 channels, which show (new Fig.3) that all three channel types are expressed by ENCCs, not just CaV3.2 as we had initially thought based on the PCR data of Hirst et al. We therefore further tested whether SAK3, the only commercially available CaV3 agonist (it amplifies CaV3.1 and CaV3.3 currents), could enhance Ca²⁺ activity. We found that it does, and added the results in Fig.3b and Fig.S10e. We further found (new Fig.4) that SAK3 could potentiate the effect of GDNF in the collagen gel migration assay, much like EDN3. We finally investigated whether SAK3 could rescue CA inhibition by BQ788 : although it did increase CA, it remained at very low levels (~0.1 events/min/100 μm²) compared to native E11.5 guts. It also did not improve ENCC migration down the colon when applied together with BQ (new Fig.5).

In light of these new results, we have shifted the focus from CaV3.2 to T-type channels in general. EWAS studies however only point to a link with CaV3.2. Tang et al. do not specify whether the variant affected channel current or expression. We reflect this in the abstract and discussion section, and, because we indeed do not have ENCC-specific KO models at this stage, nuanced our claim regarding CACNA1H and HD:

I.307: “CaV3.1 and CaV3.2 expression in E11.5 ENCCs has been previously measured by PCR²⁸; CaV3.2 was found to be ~640 times more expressed than CaV3.1. Our IHC, CA and migration assays indicate that all three CaV3 channel types are present and functional in ENCCs. The CaV3.1 & CaV3.3 agonist SAK3 promoted CA (Fig.3) and ENCC migration (Fig.4); the CaV3.2 antagonist ascorbic acid reduced CA (Fig.3) and ENCC migration (Fig.5). Very interestingly, a single-nucleotide polymorphism (SNP) of the CACNA1H gene encoding CaV3.2 has been uncovered in a recent HD exome-wide association study¹⁵. Although it is not known whether this point mutation altered CaV3.2 expression or function, our findings suggest it alters Ca²⁺ signaling. This motivates further research using dedicated mouse models to better understand the causes of neurocristopathies. We note that the CaV3.2 -/- mouse is viable^{56,57}, although the mutation induces pre-natal lethality⁵⁸. It is possible that the surviving CaV3.2 -/- embryos develop compensatory Ca²⁺ influx mechanisms, a common behavior when only one of several protein isoforms is knocked-out⁵⁹.”

o If CACNA1H loss leads to reduced ENCC migration in vivo, not just in culture.

This will be investigated in a future project with dedicated ENCC-specific knock outs of the different (or multiple) CaV3 channels.

3. Heavy Reliance on Pharmacological Inhibition Without Adequate Genetic Validation

- Most of the mechanistic claims (e.g., CaV3.2 dependence) are based on pharmacology (Z944, ascorbic acid), which:
 - o Are not entirely specific (Z944 also blocks CaV3.1).

As explained above, we have, in the light of our new IHC and pharmacological results, modified our claim from the sole involvement of CaV3.2 to that of all three CAV3 channels.

- o May affect cell health (shown for some drugs).
- No CaV3.2-specific knockdown/knockout experiments are provided to confirm specificity.
- No rescue experiments (e.g., with CaV3.2 re-expression or EDNRB overexpression) are included to test causality.

This involves genetic means which we currently do not have at our disposal.

4. Migration Assays Lack Rigorous Quantification

- The migration distances are modest and not accompanied by:

The migration distances in the cecum & colon in control conditions, 0.7-1.2 mm are in the same range as reported for 1-day culture of E11.5 guts (Hao et al. 2017), and the migration distances after 3 days in collagen gel are in the same range (0.1-0.7 mm) as reported by Bergeron et al. 2015 for identical culture conditions.

o ENCC cell number quantification—is reduced migration due to fewer cells?

o Proliferation or apoptosis assays to rule out cytotoxic effects of the drugs.

ENCC proliferation and migration are undoubtedly linked – they cannot invade the colon or the collagen gel if they do not also proliferate (“Cell proliferation drives neural crest cell invasion of the intestine”, Simpson et al., Dev. Biol., 2007). As an example, in the gel migration assay, the spherical halo measures up to 600 μm radius, whereas the initial explant typically measures only 100-200 μm . A x3-6 increase in radius means a x9-36 time increase in volume occupied by ENCCs, which can only be accounted for by proliferation. We found it more meaningful to present macroscopic migration distances (linear in the colon, spherical in the gel) rather than perform proliferation IHCs which would be more variable (because it only samples a small region of each gut) and dependent on where the section is performed. We reflected these considerations in the discussion section:

I.273: "It is likely that cytosolic Ca²⁺ oscillations are also involved in the long-term effects of EDN3/EDNRB on ENCC proliferation. ENCC proliferation and migration are inexorably linked – ENCC cannot invade the colon or a collagen gel if they do not also proliferate⁵⁴. Here, we reported macroscopic migration distances (linear in the colon, spherical in the gel) rather than local proliferation rates."

GCaMP directly allows to assess cytotoxicity because dead cells appear bright (they become permeable to Ca²⁺) and rounded. We now quantify in Fig.S16 the percentage of dead ENCCs in all chemical conditions investigated in Fig.5. The fraction of dead cells was <7% in 49/54 samples, and none of the fractions were significantly different (Mann-Whitney test and Student t-test) from the control group, although BQ788 tended to induce higher average cell death.

o In vivo rescue or dose-response studies to support pharmacological conclusions.

Z944 could be injected to pregnant mice at ENCC migration stages, however it would be difficult to rule out side-effects of this treatment, as T-type Ca²⁺ channels are expressed in many other cell types, in particular in the brain.

Considering rescue studies, we have reported in the manuscript three approaches: BQ +SAK3, BQ + 4-AP and BQ + ARL and/or ATPe. None of these approaches allowed to recover Ca²⁺ activity or migration post EDNRB-blockade by BQ788. SAK3 was however able to mimic the pro-migratory effect of EDN3 in the gel migration assay, a very interesting result (new Fig.4).

5. the rationale for doses of EDN3 added is not well explained- eg it says "1 nM EDN3 at E11.5At stage E12.5, application of 10 nM EDN3" How were the doses calculated?

We tried both 1 and 10 nM at E11.5 and at E12.5, but mentioned only the effect at 10 nM for E12.5 because it differed qualitatively from that observed at E11.5. We now include the data on 1 nM at E12.5 as well in Fig.S5c and commented in the result section:

"In stage E12.5 ileum, CA also increased threefold at 1 nM EDN3 (Fig.S5), and the application of 10 nM EDN3 induced an immediate increase of intracellular Ca²⁺ across all ENCCs (Fig.2b, Video S3, Fig.S5)."

6. some of the statements are incorrect- 'EDN3 gene knockout model that develops Hirschsprung disease, the ls/ls mouse". ls/ls is a Ednrb deficient mice not End3 knockout

ls/ls mice present a missense mutation in the EDN3 gene. We agree they are not knock-outs per-se, and have reformulated:

I.103: "We finally tested whether CA differed in an EDN3 missense mutation model that develops Hirschsprung disease ..."

I.254: "As the roles of endothelins in blood pressure regulation became more prominent, mutant mice for EDN3^{41,42} and EDNRB⁴³ mouse were re-examined"

Reviewer #3 (Remarks on code availability):

There is no code involved